# Multisensor validation of tidewater glacier flow fields derived from SAR intensity tracking

Christoph Rohner[1], David Small[1], Daniel Henke[1], Martin P. Lüthi[1], and Andreas Vieli[1]

[1]Department of Geography, University of Zurich, CH-8057 Zurich, Switzerland

**Correspondence:** Christoph Rohner (christoph.rohner@geo.uzh.ch)

**Abstract.** Following the general warming trend in Greenland, an increase in calving rates, retreat and ice flow has been observed at ocean-terminating outlet glaciers. These changes contribute substantially to the current mass loss of the Greenland Ice Sheet. In order to constrain models of ice dynamics as well as estimates of mass change, detailed knowledge of geometry and ice-flow are needed, in particular on the rapidly changing tongues of ocean-terminating outlet glaciers. In this study, we validate velocity estimates and spatial patterns close to the calving terminus of such an outlet derived from an iterative offset tracking method based on SAR intensity data with a collection of three independent reference measurements of glacier flow. These reference data sets are comprised of measurements from differential GPS, a Terrestrial Radar Interferometer (TRI) and repeated UAV surveys. Our approach for the SAR-velocity processing aims at achieving a relatively fine grid spacing and a high temporal resolution in order to best resolve the steep velocity gradients in the terminus area and to exploit the 12 day repeat interval of the single-satellite Sentinel-1A sensor. Results from images of the medium-sized ocean terminating outlet glacier *Eqip Sermia* acquired by Sentinel-1A and RADARSAT-2 exhibit a mean difference of 11.5% when compared to the corresponding GPS measurements. An areal comparison of our SAR velocity-fields with independently generated velocity maps from TRI and UAV showed a good agreement in magnitude and spatial patterns, with mean differences smaller than 0.7 md[-1]. In comparison with existing operational velocity products, our SAR-derived velocities showed an improved spatial velocity pattern near the margins and calving front. There 8% to 30% higher surface ice velocities are produced, which has implications on ice fluxes and on mass budget estimates of similar sized outlet glaciers. Further, we showed that offset tracking from SAR intensity data at relatively low spatio-temporal sampling intervals is a valid method to derive glacier flow fields for fast-flowing glacier termini of outlet glaciers and, given the repeat period of 12 days of the Sentinel-1A sensor (6 days with Sentinel-1B), has the potential to be applied operationally in a quasi-continuous mode.

## 1 Introduction

As a result of the general warming trend in Greenland and the migration of subtropical water currents toward Greenland's coast, ice loss by submarine melt and iceberg calving – a process neither well understood nor well represented in the current generation of ice-sheet models – is increasing (Straneo et al., 2013). The related dynamic mass loss is expected to further intensify in the future, thereby strongly contributing to global sea level rise (IPCC, 2013; Nick et al., 2013). The increase in calving activity is related to substantial terminus retreat, thinning and speed-up. Over the past two decades, such flow acceleration has

exceeded 30% for many of the Greenland Ice Sheet's (GrIS) outlet glaciers in the northeast and southwest (Rignot et al., 2008; Moon et al., 2012; Wood et al., 2018). Regarding the future of the Greenland Ice Sheet in context of climate change, detailed and repeated observations of flow velocities of tidewater outlet glaciers are crucial for assessing the mass budget of the GrIS, for better understanding the mechanisms behind dynamic mass loss and for developing and constraining predictive flow models
(Vieli and Nick, 2011).

Due to the remoteness of the Arctic region, *in situ* measurements are not only expensive and logistically difficult, but also limited in spatial and temporal coverage (Joughin, 2002; Euillades et al., 2016). In addition, the operational use of optical remote sensing to measure flow dynamics is limited by the availability of sunlight during the long polar winter as well as cloud cover. The launch of Sentinel-1A (S-1A) by the European Space Agency (ESA) in 2014 and Sentinel-1B (S-1B) in 2016 drastically
increased the availability of active remote sensing data (i.e. Synthetic Aperture Radar, SAR). Independent of the availability of sunlight and unaffected by cloud cover, SAR systems are able to circumvent the aforementioned drawbacks of passive (optical) systems, allowing for spatially and temporally quasi-continuous measurements of the ice sheets' flow dynamics as well as many other cryospheric parameters (Joughin et al., 2016; Dowdeswell et al., 1999). Our velocity processing is therefore focused on SAR imagery and specifically on the Sentinel-1 sensor, as a 12 day repeat image acquisition of relatively high
spatial resolution (2.3×14.1 m) is already operational (6 days with S-1B; cf. Sect. 2.1).

Using SAR systems, flow velocities can be estimated using either repeat-pass interferometry (InSAR) or offset tracking approaches. As reported by Michel and Rignot (1999) and Joughin (2002), fast moving glaciers in combination with relatively long repeat cycles can cause difficulties concerning the maintenance of coherence, limiting the InSAR-based velocity estimation to slow-moving areas, reducing the use of this approach close to the glaciers' termini. Furthermore, the presence of
surface melt and high strain rates near the glacier's terminus reduces the usability of InSAR for velocity estimation. The offset-tracking methodology laid out by Scambos et al. (1992) and Frezzotti et al. (1998) for optical data and by Fahnestock et al. (1993), Gray et al. (1998), Michel and Rignot (1999) and Joughin (2002) using SAR imagery offers an alternative approach for velocity estimation. Relying on cross-correlating the speckle patterns of an image pair, offset-tracking is not as sensitive to decorrelation as InSAR and therefore allows derivation of flow dynamic parameters even in faster moving parts of the glacier
(Joughin, 2002; Luckman and Murray, 2005). Despite the widespread use of SAR datasets for derivation of flow velocities (e.g. Gray et al., 2001; Lemos et al., 2018), fewer studies have been devoted to analysing the accuracy in magnitude and spatial patterns of the derived flow velocity products compared to in-situ measurements, in particular close to the calving terminus (e.g. Nagler et al., 2015; Joughin et al., 2018). In view of the use of such SAR-derived velocities for mass change assessments, calving-process studies or as model constraints, a comprehensive evaluation of the performance, uncertainties and drawbacks
of this spaceborne method is crucial.

Of the number of articles analysing the validity of velocity products derived from SAR sensors, almost all focus on results using the interferometric approach (e.g. Goldstein et al., 1993; Rignot et al., 1995; Gray et al., 1998), therefore limited to slow-moving areas. With respect to the use of intensity tracking to derive flow velocities, only a small number of articles validated the results against *in situ* measurements (e.g. Fallourd et al., 2011; Schubert et al., 2013; Schellenberger et al., 2015), all focus-
ing on glaciers reaching a maximum of <3 md$^{-1}$. The works of Ahlstrøm et al. (2013, ALOS/PALSAR, TerraSAR-X/Tandem-X

data) and Boncori et al. (2018, ALOS/PALSAR, ASAR, and ERS-1/-2 data) looked at both, interferometric and offset tracking approaches in combination with GPS measurements.

In this paper, we are investigating the limits of offset tracking methods and demonstrate that accurate estimates are possible even close to the calving front when choosing appropriate template sizes. As validation datasets, for the derived flow velocity
information (magnitude and spatial pattern) we use field measurements based on three independent methods, namely differential GPS (dGPS), a terrestrial radar interferometer (TRI), and high resolution imagery from an Unmanned Aerial Vehicle (UAV). The glacier studied is *Eqip Sermia*, a medium sized marine-terminating outlet glacier in the southwest of Greenland (cf. Fig. 1). We demonstrate that flow velocity estimates generated at a relatively fine ground sampling distance are more accurate close to the glacier's terminus compared to operational, ice-sheet wide ice velocity products which tend to underestimate
the glacier's dynamics and thus also the calving flux.

## 1.1   Study Area

For this study, the medium-sized ocean terminating outlet glacier of *Eqip Sermia* (69°48' N, 50°13' W), located in Western Greenland, was observed by multiple sensors. Given the available historical geometry and flow velocity survey data, dating back to 1912 (e.g. de Quervain and Mercanton, 1925), *Eqip Sermia* offers ideal preconditions. *Eqip Sermia* has a calving front
roughly 3.5 km wide and 30–200 m high. The long-term flow speed at the terminus was stable for almost a century at about 3 md$^{-1}$ (Bauer, 1968), followed by an acceleration towards the end of the 20th century. Between 2000 and 2005, *Eqip Sermia* accelerated by 30% as well, doubling the discharge (Rignot and Kanagaratnam, 2006; Kadded and Moreau, 2013; Lüthi et al., 2016). More detailed velocity fields from the recent decade (Joughin et al., 2008, 2010) indicate strong spatial variations in flow in the terminus area and a strong acceleration towards the calving front (Lüthi et al., 2016; Catania et al., 2018).

## 2   Data and Methods

## 2.1   SAR Data

The increasing availability of freely and openly available SAR data over the ice sheets at high spatial and temporal resolutions allows circumvention of acquisition issues intrinsic to optical systems that have been available for several decades. Making use of the all-weather, day/night imaging capabilities of SAR sensors, cloud cover or illumination effects do not interfere with
the acquisition schedule. The source data consists of a dual-pol (HH/HV) Sentinel-1A/B Single Look Complex (SLC) C-band (5.405 GHz) time-series starting in October 2014, accessed through the Copernicus Open Access Hub (Torres et al., 2012). Out of these more than 200 acquisitions, 5 scenes complement ground measurement data acquired during a field campaign that took place in August 2016. These S-1A satellite products have a 12-day repeat cycle and all interferometric wide swath (IW-mode) SLC products used were acquired from the same ascending relative orbit (cf. Table 1). After this validation campaign, the
S-1B satellite was commissioned, lowering the S-1 repeat interval to 6 days. In addition to the S-1A data sets, a total of 20 RADARSAT-2 (RS-2) acquisitions were made available through the Science and Operational Applications Research Program

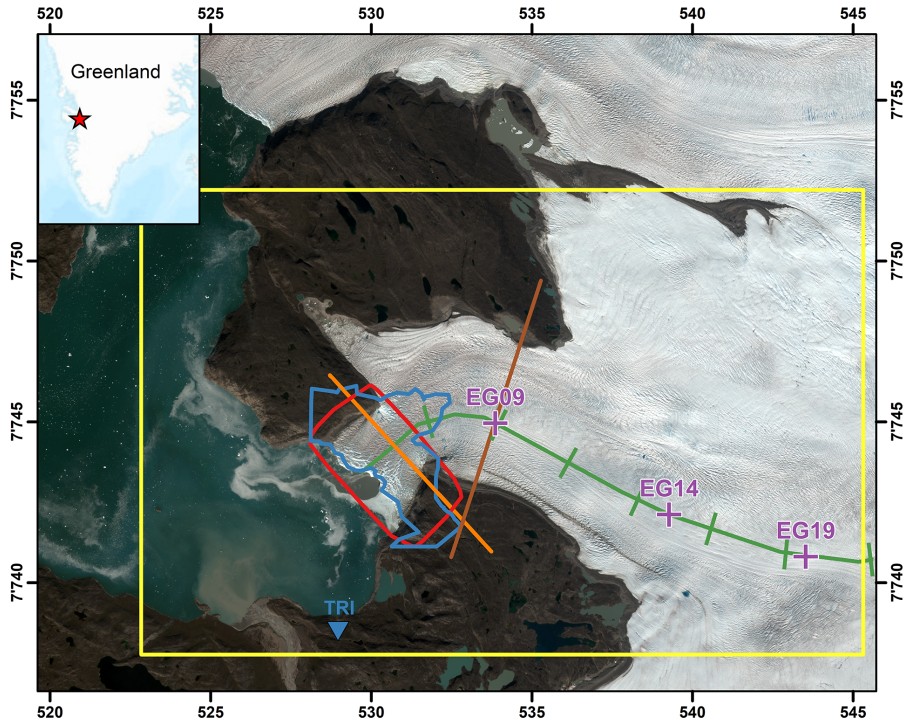

**Figure 1.** Eqip Sermia Glacier in West Greenland. The yellow area indicates the region evaluated in this study using spaceborne SAR images (S-1A and RADARSAT-2), with the red line showing the extent of the mosaic built up using airborne UAV and the blue polygon depicting the area with mean ground-based backscatter coherence >0.7 acquired from the terrestrial radar interferometer (TRI; its position is indicated with the blue triangle). The purple crosses mark the GPS tracker positions. In orange and brown the flowlines across the tongue are depicted (cf. Fig. 14 and 15). The green line depicts the central flowline with distance markers every 2500 m, starting at the glacier's terminus (cf. Fig. 13). Background: Sentinel-2A scene from 3 August 2016 (UTM 22N projection). Contains Copernicus Sentinel data (2016).

(Project CSA-SOAR-EU-16821). Based on this allowance, two SLC scenes were acquired using RS-2's Ultra-Fine wide mode operating also at the frequency of 5.405 GHz with a temporal baseline of 24 days (cf. Table 1). The detected HH polarized SAR images from both sensors were geometrically terrain corrected using Range-Doppler geocoding (Meier et al., 1993) based on the "GIMP" Digital Elevation Model (DEM). The Greenland Ice Mapping Project (GIMP) DEM has a grid spacing of $30 \times 30$ m (Version 2.1; Howat et al., 2014), which was oversampled to $2.5 \times 2.5$ m. As we operated in the DEM geometry, no separate co-registration was performed. No tiepoints were employed during geocoding, as the geolocation accuracy was sufficient (Schubert et al., 2017). The HH polarization was chosen due to its higher signal-to-noise ratio.

## 2.2 SAR Intensity Tracking

With the increasing availability of space-borne interferometric data and using well-established methodologies, measurements of ice-velocities at high spatio-temporal resolutions and accuracies of a few meters per year are performed on a regular basis for

**Table 1.** Sentinel-1A/RADARSAT-2 acquisitions of *Eqip Sermia* investigated in this study

| Acquisition Date | Platform | Mode | Product Type | Orbit Number | Pass Direction | Pixel Spacing (rg × az) |
|---|---|---|---|---|---|---|
| 2016/07/08 | S-1A | IW | SLC | 12062 | Ascending | 2.3×14.1 m |
| 2016/07/20 | S-1A | IW | SLC | 12237 | Ascending | 2.3×14.1 m |
| 2016/08/01 | S-1A | IW | SLC | 12412 | Ascending | 2.3×14.1 m |
| 2016/08/06 | RS2 | Ultra-Fine Wide | SLC | 311 | Ascending | 1.3×2.1 m |
| 2016/08/13 | S-1A | IW | SLC | 12587 | Ascending | 2.3×14.1 m |
| 2016/08/25 | S-1A | IW | SLC | 12762 | Ascending | 2.3×14.1 m |
| 2016/08/30 | RS2 | Ultra-Fine Wide | SLC | 311 | Ascending | 1.3×2.1 m |

the polar regions. Glacier flow speeds as low as $200\,\mathrm{ma^{-1}}$ have been reported as a practical upper limit for phase interferometry with a 24 day repeat-cycle duration (Gray et al., 1998; Joughin, 2002). As this study focuses on the glacier's flow dynamics at the fast-moving glacier terminus area, interferometric methods were not expected to be viable with a temporal baseline of 12

days. In addition to the temporal decorrelation, InSAR is limited to observing the line of sight. By evaluating both ascending and descending tracks (Joughin et al., 1998) and making an assumption of surface-parallel flow of the surface ice one can retrieve a full 3D displacement using two different tracks. As only ascending geometries were available in this instance for *Eqip Sermia*, we used a speckle tracking approach to derive the glacier's movements. This method makes use of the backscattered speckle pattern within image patches from subsequent, co-registered image acquisitions to derive two-dimensional offset values

by calculating the normalized cross-correlation between the image patches (Gray et al., 2001; Strozzi et al., 2002; Joughin, 2002). As this approach does not rely on phase information, using instead the detected SAR image, phase decorrelation caused by meteorological conditions or incoherent and/or rapid flow does not influence the velocity estimation and therefore allows retrievals at higher ice speeds and longer orbit repeat intervals (Gray et al., 2001; Strozzi et al., 2002). Nevertheless, strong changes in the amplitude of the backscattered signal (e.g. due to substantial changes in the presence of surface melt water) may

result in a deterioration of the velocity estimates.

Following Strozzi et al. (2002), the methodology outlined in Fig. 2 was implemented in MATLAB, resulting in pixel-wise X- and Y-offsets. A patch size of $101\times101$ pixels was chosen for the template image, corresponding to about $250\times250\,\mathrm{m}$. Given the glacier's flow velocity of up to $15\,\mathrm{md^{-1}}$ at the glacier front (Lüthi et al., 2016) and the temporal baseline of 12 days between repeat-pass Sentinel-1A acquisitions (24 days for RS-2), the search region was set to $181\times181$ pixels for S-1A IW

product pairs and $261\times261$ pixels for RS-2. Each of the resulting correlation matrices was oversampled by a factor of 9 in both dimensions to locate the sub-pixel position of the correlation peak. This procedure was repeated for every second pixel in both dimensions, resulting in a sample interval of the initial velocity map ($V_{map}$) of $5\times5\,\mathrm{m}$. Choosing a smaller sample step size is beneficial for capturing of strong velocity gradients (e.g. close to the glacier's terminus), but at an increased computational cost and a known correlation between adjacent pixels due to overlap of subsequent image patches.

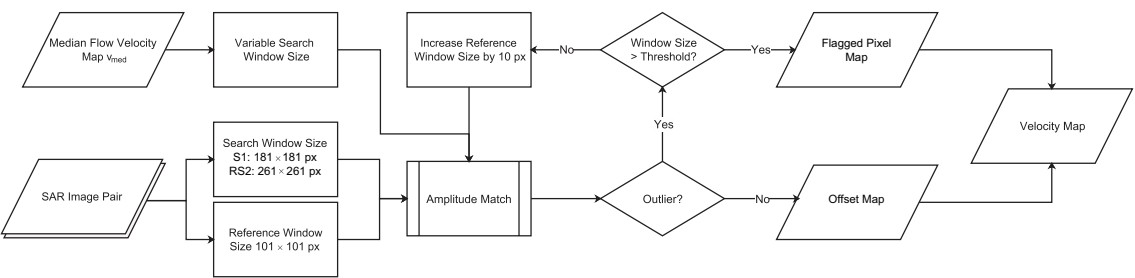

**Figure 2.** Amplitude Match Methodology for 12-day Sentinel-1A SAR products. The same methodology was used for RADARSAT-2 image pairs, but with an increased search window size of 261×261 pixels to accommodate the 24-day repeat orbit.

**Table 2.** Overview of search window sizes used to calculate offsets for the long-term flow velocity average product ($V_{med}$). Image pairs with a temporal baseline >24 days (in rare cases of extended missing acquisitions) were not used.

| Temporal baseline | Search Window Size (pixels) | Search Window Size (m) |
|:---:|:---:|:---:|
| 6 days | 141×141 | 352.5×352.5 |
| 12 days | 181×181 | 452.5×452.5 |
| 24 days | 261×261 | 652.5×652.5 |

### 2.2.1 Outlier Detection and Process Iteration

To cull out bad matches, the two-step approach outlined in Fig. 3 was implemented, using a long-term flow velocity average product ($V_{med}$) and a correlation value threshold. For this $V_{med}$, offsets in X- and Y-direction were calculated using a 101×101 pixel (252.5×252.5 m) template patch size and search window sizes based on the temporal baseline of each of the 256 image pairs available between 2014/10/11 and 2018/03/18 (cf. Table 2). Subsequently, a median flow velocity ($V_{med}$) and median flow angle map ($\angle_{med}$) were computed based on these 256 flow fields. The correlation signal-to-noise-ratio (SNR) was calculated by dividing the maximum height of the correlation peak ($C_{max}$) by the mean level of the correlation function ($C_{mean}$) for every pixel (Strozzi et al., 2002).

An outlier mask was calculated following Eq. (1).

$$(V_{map} > 2 \cdot V_{med}) \vee (SNR < 2 \wedge C_{max} < 0.5) \vee \angle_{diff} > 45°, \tag{1}$$

where $\angle_{diff} = |\angle_{med} - \angle_{map}|$ and $\angle_{map}$ representing the mapped flow angle from the image pair. For every pixel flagged as an outlier, the intensity tracking was reiterated with the reference patch size increased by 10 pixels in each dimension. The patch size of the search region was defined for every outlier pixel as a function of $V_{med}$, limiting the search region to three times the expected displacement, thus decreasing the chance of missing the peak correlation. Following the outlier detection, iterative intensity tracking for outlier pixels was repeated for a maximum of five times or until no further outlier pixels were detected. The application of the iteration procedure drastically reduced the number of void pixels, while enabling reasonably

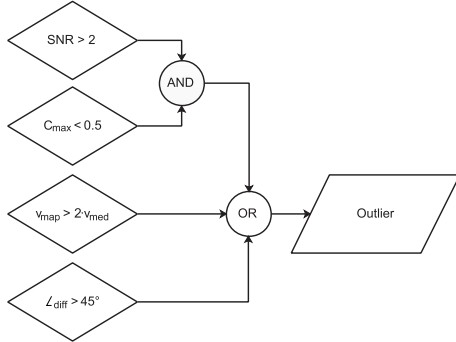

**Figure 3.** Two-step outlier detection approach, with $C_{max}$ being the maximum, $C_{mean}$ the absolute mean value for each correlation function. The signal-to-noise ratio (SNR) is calculated by dividing $C_{max}/C_{mean}$. $V_{map}$ describes the calculated flow velocity for an image pair, $V_{med}$ the long-time median.

small patch sizes and therefore meaningful results even close to the lateral glacier margin and in particular at the calving front. Following the outlier detection and process iteration step, $V_{map}$ was downscaled to a pixel spacing of $100 \times 100$ m by applying a median filter to account for the spatial correlation of adjacent pixels caused by overlapping template patches in the initial $V_{map}$ (cf. Sect. 2.2).

### 2.3 GPS Velocity Data

For validation purposes, seven low-cost single-frequency continuous GPS receivers (Wirz et al., 2013; Buchli et al., 2012) were deployed on the glacier using a photovoltaic system in combination with a battery (cf. Fig. 1). In addition to the receivers on the glacier, a base station was deployed on bedrock. The devices were installed on June 29, 2016: five were recovered by August 25, 2016, a sixth one was recovered in Summer 2017 and GPS solutions at 24 h intervals were calculated at the Geodesy and Geodynamics Lab of ETH Zurich (Wirz et al., 2013). To derive robust average velocities over the relevant image-pair periods, the 24 h GPS solutions were further processed using a three-step approach that begins with the culling of outliers, followed by temporally averaging both, the latitudinal and longitudinal positions as well as the resulting velocities, in the manner described by Ahlstrøm et al. (2013). The velocities of the three sensors located in parts of the glacier with flow velocities higher than 1 md$^{-1}$ (GPS sensors EG09, EG14 and EG19) were chosen for accuracy assessment, as lower flow velocities exhibit relatively low signal-to-noise ratios (SNR) when intensity tracking was applied to two immediately successive S-1A SAR acquisitions.

### 2.4 UAV survey

To acquire additional ground-truth data, an unmanned aerial vehicle (UAV) was flown over *Eqip Sermia* on three occasions in August 2016 (cf. Table 3). The UAV surveys were carried out using a SenseFly eBee system, a light, fixed-wing UAV with a wingspan of 96 cm (senseFly SA, 2017). The images were acquired using a modified Sony Cyber-shot DSC-WX220 with 18.2 MP and a sensor size of $6.17 \times 4.55$ mm. For every image, the approximate 3D position as well as the UAV's orientation

**Table 3.** Overview of acquisitions using the UAV during the 2016 field campaign. Date of acquisition is listed together with the average ground sampling distance (GSD), number of images acquired ($N_{img}$) and the area covered.

| Name | Date | GSD | $N_{img}$ | Area covered | Weather | Remarks |
|------|------|-----|-----------|--------------|---------|---------|
| eBee1 | 2016/08/21 | 17.21 cm | 628 | 12.87 km$^2$ | Clear Skies | |
| eBee2 | 2016/08/23 | 17.93 cm | 214 | 7.06 km$^2$ | Overcast | Acquisition aborted due to high winds |
| eBee3 | 2016/08/25 | 17.3 cm | 616 | 12.53 km$^2$ | Overcast | |

(i.e. roll, pitch, and yaw) were annotated based on information from the on-board GPS and inertial measurement unit (IMU) (senseFly SA, 2016, 2017). A total of 6 flights per mission were carried out in parallel strips within 3 h, resulting in an average of more than 5 overlapping images (i.e. 70% longitudinal/lateral overlap per image), with reduced redundancy near the outer limits of the acquisition areas. For each flight, the images acquired by the eBee UAV were processed by applying the structure from motion technique using Pix4Dmapper Pro (Eltner and Schneider, 2015), resulting in an optical mosaic (RGB) and a digital surface model (DSM) of the area with a ground sampling distance (GSD) of ∼0.17 m (cf. Table 3). The resulting datasets were georeferenced using 5 manually surveyed ground control points (GCP). As it was not possible to reach the northern lateral bedrock due to heavy crevassing, these GCPs were restricted to one side of the glacier, resulting in small residual differences between the geolocation accuracy of some mosaics. To reduce differences between the mosaics, an affine transformation based on a total of 40 manually selected GCPs was performed with acquisition eBee1 (cf. Table 3) as a master reference and the other two acquisitions as slaves.

Due to differences in illumination between the acquisitions eBee1 and eBee3 (cf. Table 3), image matching using the different optical bands was not feasible. We therefore used the DSMs to derive shaded reliefs and input these to the image matching algorithm, to minimize matching errors caused by illumination differences between the data sets. Flow velocity and direction were estimated by applying the approach described in Sect. 2.2. Mosaics eBee1 and eBee3 with a temporal baseline of 4 days (cf. Table 3) were chosen, as only those two included the tongue's full extent. Despite the difference in pixel spacing between the SAR images and the UAV's mosaics, the reference window size remained the same (101×101 pixels, ∼17×17 m), together with a larger search window size of 501×501 pixels (∼85×85 m) to accommodate the glacier's movement during the 4 days between the acquisitions.

## 2.5  Terrestrial Radar Interferometer Data

In order to reference flow velocity and geometry information with high spatial and temporal resolution at the calving front, a terrestrial radar interferometer (TRI; Caduff et al., 2015) was set up on stable ground 5 km south of the calving front with an unobstructed view of the glacier. The TRI system used was a GAMMA Portable Radar Interferometer (GPRI; Werner et al., 2008), operating at Ku-band (17.2 GHz). The device operates as a real-aperture radar interferometer, having one transmitting and two receiving antennas. The glacier was scanned at 1 min intervals for 8 consecutive days. Occasional data gaps were caused by hard-drive issues. The resulting radar intensity and phase measurements were processed further by application of

a standard workflow to determine the displacements in line-of-sight (e.g. Caduff et al., 2015; Lüthi et al., 2016). According to Voytenko et al. (2015) the absolute velocity errors were <0.5 md$^{-1}$ with averaging times of ~1 hour even for distant points in a humid atmosphere, with a range resolution of 0.75 m and a linearly scaling azimuth resolution of 35 m at 5 km distance (Werner et al., 2008).

## 3 Results

A typical example of a velocity field derived from a 12 day Sentinel-1A repeat acquisition in August 2016 is shown in Fig. 4. In the main upstream tributary trunk of the glacier flow speeds were in between 1 and 2.5 md$^{-1}$, decreasing to zero towards the lateral margins. Within the last 5 km towards the calving front, the flow in the center line strongly increases to maximum values reaching up to 7 md$^{-1}$ but with a rapid decrease towards the lateral margins. Within the last few hundred meters of the calving front, velocities dropped again substantially due to image templates containing mixed information from the glacier and the sea (or ice-mélange in winter), leading to erroneous underestimations.

Note that the velocity field in Fig. 4 is from one single 12 day pair and was only filtered for outliers (magnitude and direction) and downscaled to 100×100 m but no further smoothing was applied, which explains the data voids and somewhat noisy appearance.

### 3.1 Comparison of GPS data to satellite-derived velocities

Using the GPS based flow velocity measurements as a first ground control dataset, we were able to assess the accuracy of the offset tracking approach applied to satellite products. In a first comparison, the flow velocities derived from spaceborne SAR satellites (cf. Table 1, example in Fig. 4) were plotted against the mean GPS flow speeds of the corresponding periods, as shown in Fig. 5. A total of 13 SAR derived velocity estimates were compared with the GPS derived flow velocities, five periods for GPS tracker EG09 and four periods each for GPS trackers EG14 and EG19 (cf. Table 4). To improve the SNR of the intensity tracking estimates, the integration time of the velocity estimates based on S-1A was doubled to 24 days for the two GPS stations located in the upper part of the glacier (i.e. EG14 and EG19) where the velocities are relatively low (1-1.5 md$^{-1}$). In addition to the improvement in SNR and thus a decrease in pixels flagged as outliers (cf. Fig. 4), this allowed for direct comparisons with results from RS-2 which also had a 24 day orbital repeat. The GPS velocities were averaged over the same periods as the satellite observation interval. Comparing the flow velocities, they showed generally good agreement (R$^2$: 0.674), with a mean relative difference of 11.5% between the two methods and an RMSE of 0.202 md$^{-1}$. When comparing the average of both, the GPS measurements and SAR derived flow velocities, for the whole campaign duration (about two months), the mean relative differences were reduced to 9.1% for EG09, 8.4% for EG14, and 5.4% for EG19.

### 3.2 Comparison of satellite-derived to UAV-derived velocities

The UAV-derived velocity field (2016/08/21 and 2016/08/25, Table 3), shown in Fig. 6, covered the lower fast flowing 5 km of the glacier tongue and in general showed a very similar spatial pattern in flow speed to the SAR-derived velocities. The

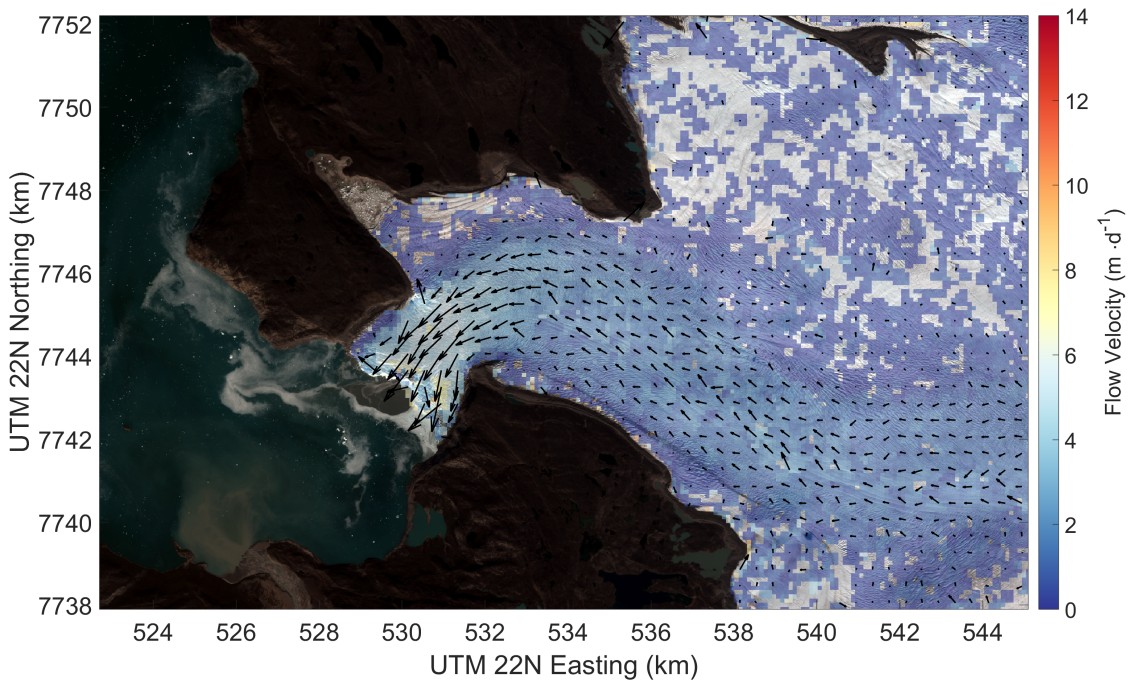

**Figure 4.** Flow velocity field (magnitude and direction) from Intensity Tracking based on subsequent Sentinel-1A acquisitions with a time difference of 12 days (2016/08/13 and 2016/08/25) from Intensity Tracking. Note that the low values close to the calving margin are due to image templates containing mixed information from the glacier and the sea, leading to erroneous values. Transparent areas on the glacier depict pixels flagged as outliers. Background: Sentinel-2A scene from 3 August 2016. Contains Copernicus Sentinel data (2016).

main differences were that the flow field was much smoother and that it was additionally able to significantly better resolve the acceleration towards the terminus where flow speeds of $12\,\mathrm{md^{-1}}$ are reached. These discrepancies can be attributed to the
15  much higher spatial resolution and hence differences in patch size. This improved spatial resolution even allowed resolution of discontinuities in flow speed near the calving front related to deep crevasses and rifts. The UAV-derived flow field (Fig. 6) also confirms the decreased but non-zero flow along the orographic left margin on the tongue already indicated in the SAR data (Fig. 4).

In order to quantitatively (pixel by pixel) compare the flow speeds from the SAR with the UAV, the resolution differences between the radar and eBee datasets needed to be resolved. Therefore, the flow velocity map based on the UAV data was downscaled to match the $100\,\mathrm{m}$ grid size of the SAR based result. A mask was then applied for the areas outside of the glacier
tongue to exclude stationary (e.g. moraine) and incoherently moving areas (e.g. open water) from the statistics (cf. black outline in Fig. 7). The mask was manually traced using a Sentinel-2 scene acquired on 2016/08/03 as a reference.

The comparison between the two methods showed generally good agreement in most parts of the glacier tongue with differences between $\pm 1\,\mathrm{md^{-1}}$. However, a discrepancy was observed near the calving front as well as at the side of the glacier, where

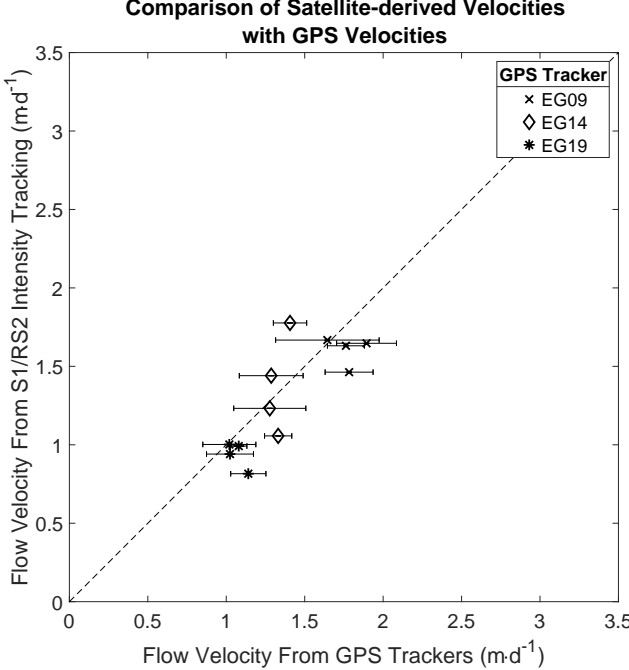

**Figure 5.** Comparison of satellite-derived velocities with GPS velocities. To account for low SNR values in areas with flow velocities $\sim 1\,\text{md}^{-1}$ (EG14/EG19), integration time was doubled to 24 days. Vertical bars depict the standard deviation of flow speeds based on Intensity Tracking, horizontal bars show the standard deviation of the GPS velocities.

differences exceeded $\pm 2\,\text{md}^{-1}$. Due to these larger differences, the standard deviation was $1.915\,\text{md}^{-1}$ with a mean difference of -0.689 and a median of -0.281, showing a shift towards higher flow velocities based on the UAV data. These differences were expected, as the initial $101 \times 101$ pixels (i.e. $252.5 \times 252.5$ m) template size crossed the glacier's boundaries at the calving front as well as at the sides, resulting in miscorrelations and therefore incorrect flow speeds from the SAR data, in particular close to the calving front. When possible border regions were excluded using a 250 m buffer around the glacier mask, the statistical values improved marginally (cf. red line in Fig. 7), resulting in a standard deviation of $1.576\,\text{md}^{-1}$ and mean and median values of -0.626 and $-0.317\,\text{md}^{-1}$ respectively.

One should note that the time-periods of data acquisition were not identical (4 days UAV vs. 12 days SAR), but temporal variations within these time-scales reached their minimum at upstream GPS EG09 and were only substatial within 500 m of the terminus in the continuous TRI data (personal communication Andrea Walter, 26 Oct 2018).

## 3.3 Comparison of UAV-derived to interferometrically measured velocities

The TRI derived velocity map of the lower tongue was not continuous due to shadows from topography with respect to the TRI line of sight. The general spatial patterns were very similar to the UAV data, confirming the spatial gradients towards the lateral margins as well as the strong rapid step-wise acceleration within the last 1 km towards the front. The flow at the calving

**Table 4.** Comparison of Flow Velocities derived by Intensity Tracking from Sentinel-1/RADARSAT-2 radar imagery and from GPS measurements. Ice velocities are given in md$^{-1}$, UTM 22N coordinates in m.

| Tracker | Platform | Start Date | End Date | Mean UTM 22N Coordinates | | vGPS | Std GPS | vSAR | vGPS-vSAR |
|---------|----------|------------|----------|------------|------------|------|---------|------|-----------|
| EG09 | RS2  | 2016/08/06 | 2016/08/30 | 533758.34 | 7744997.41 | 1.70 | 0.28 | 1.68 | 0.02 |
| EG09 | S-1A | 2016/07/08 | 2016/07/20 | 533813.03 | 7744977.74 | 1.89 | 0.19 | 1.65 | 0.24 |
| EG09 | S-1A | 2016/07/20 | 2016/08/01 | 533792.29 | 7744985.37 | 1.78 | 0.15 | 1.46 | 0.32 |
| EG09 | S-1A | 2016/08/01 | 2016/08/13 | 533772.48 | 7744992.40 | 1.76 | 0.12 | 1.63 | 0.13 |
| EG09 | S-1A | 2016/08/13 | 2016/08/25 | 533752.48 | 7744999.50 | 1.64 | 0.33 | 1.67 | -0.03 |
| EG14 | RS2  | 2016/08/06 | 2016/08/30 | 539286.31 | 7742106.13 | 1.28 | 0.23 | 1.23 | 0.05 |
| EG14 | S-1A | 2016/07/08 | 2016/08/01 | 539317.98 | 7742089.59 | 1.41 | 0.11 | 1.78 | -0.37 |
| EG14 | S-1A | 2016/07/20 | 2016/08/13 | 539303.35 | 7742097.21 | 1.33 | 0.09 | 1.06 | 0.27 |
| EG14 | S-1A | 2016/08/01 | 2016/08/25 | 539289.24 | 7742104.61 | 1.29 | 0.20 | 1.44 | -0.15 |
| EG19 | RS2  | 2016/08/06 | 2016/08/30 | 543547.47 | 7740771.73 | 1.02 | 0.17 | 1.00 | 0.02 |
| EG19 | S-1A | 2016/07/08 | 2016/08/01 | 543576.17 | 7740773.70 | 1.14 | 0.11 | 0.82 | 0.32 |
| EG19 | S-1A | 2016/07/20 | 2016/08/13 | 543562.81 | 7740772.75 | 1.08 | 0.05 | 0.99 | 0.09 |
| EG19 | S-1A | 2016/08/01 | 2016/08/25 | 543550.13 | 7740771.90 | 1.02 | 0.15 | 0.94 | 0.08 |

front is now also very well resolved and there maximum speeds in line of sight of 14 md$^{-1}$ are reached.

Again for a detailed quantitative comparison of velocity measurements from the TRI measured interferometrically with those derived from the UAV's hillshaded DSMs, the UAV-derived velocity field was first projected into the line-of-sight direction relative to the TRI's position, resampling the data onto the same grid. The comparison of these projected estimates and the interferometrically derived TRI flow velocities (cf. Fig. 9) show a close correspondence of the data for all areas aside from the very front and the lateral edges of the UAV's acquired area. The frontal differences can be explained by the differences in the

measurement techniques, measurement times and spatial resolutions used. The image matching algorithm used with the UAV data relies on features visible in both the reference and search scenes. Due to calving events, features at the glacier front may no longer exist in subsequent acquisitions, causing phantom image correlations. As the TRI relies on direct interferometric measurements at 1 minute intervals, it is less susceptible to errors caused by calving. As mentioned above, the point density of the drone's DSM was reduced at the margins of the acquired area, resulting in error-prone image matching results in those areas. Note that the non-zero ice flow at the orographic left lateral margin (in contrast to the right margin) was again very well reproduced by both acquisition methods.

Inspecting the overlapping areas between the two sensors within the glacier's marginal boundary (black line in Fig. 9), eBee showed a mean velocity of 1.982 md$^{-1}$ vs. 2.62 md$^{-1}$ for the TRI. The mean difference between the two datasets was 0.633 md$^{-1}$; the median difference was -0.007 md$^{-1}$ (cf. Fig. 9). The standard deviation was 2.9 md$^{-1}$. After applying a 250 m buffer around the glacier's margin (cf. red line in Fig 9), the statistical values changed to a standard deviation of 1.844 md$^{-1}$ and mean and median differences of 0.372 and 0.129 md$^{-1}$ respectively.

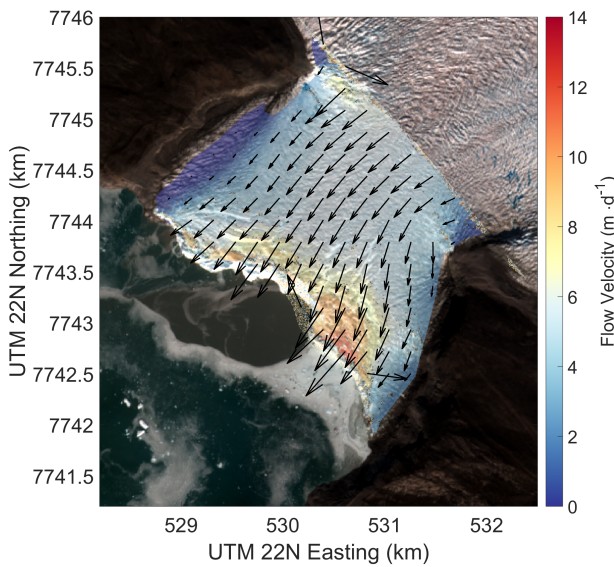

**Figure 6.** Flow velocity field derived from Intensity Tracking using subsequent UAV acquisitions with a time difference of 4 days (2016/08/21 and 2016/08/25). Due to illumination differences between the acquisitions, shaded surface reliefs derived from structure from motion were used for the image matching procedure. Note the jumps in glacier flow speed close to the calving front that can be attributed to the presence of rifts. Erroneous values close to the calving margin can be caused by image templates containing mixed information from the glacier and the sea. Background: Sentinel-2A scene from 3 August 2016. Contains Copernicus Sentinel data (2016).

## 4   Discussion

The results and comparisons to other data presented here show the feasibility but also the limitations of operational offset tracking using Sentinel-1 intensity data to estimate glacier flow dynamics at relatively high spatial ($100 \times 100$ m) and temporal sample intervals (12 to 24 days). While the applicability of interferometric approaches has been demonstrated in the past, its applicability is limited by temporal decorrelation in cases of fast movements which is particularly the case on tongues of tide-water outlet glaciers. The S-1/GPS comparison highlights the validity of the feature tracking approach with a mean velocity difference of 11.5% between the datasets, agreeing with the reported differences of 9.7% by Ahlstrøm et al. (2013). Further, in comparison to the UAV-derived velocities (which agrees with TRI), our approach is able to represent the spatial pattern of ice flow towards the fast-flowing calving front well. Specifically, the main acceleration in flow towards the front as well as gradients to the margins are well reproduced. However, some edge effects mostly at the calving front remain, due to overlaps of the templates used in the cross-correlation with non-glaciated areas, but this effect is limited to a narrow zone determined by the chosen template dimension and is mostly filtered out when considering flow directions as well. The non-zero velocity at the orographic left side of the terminus is reproduced surprisingly well by our 12 day SAR estimates, indicating that the inability of deducing strong velocity gradients within template patches mentioned by Nagler et al. (2015) does not impact our velocity

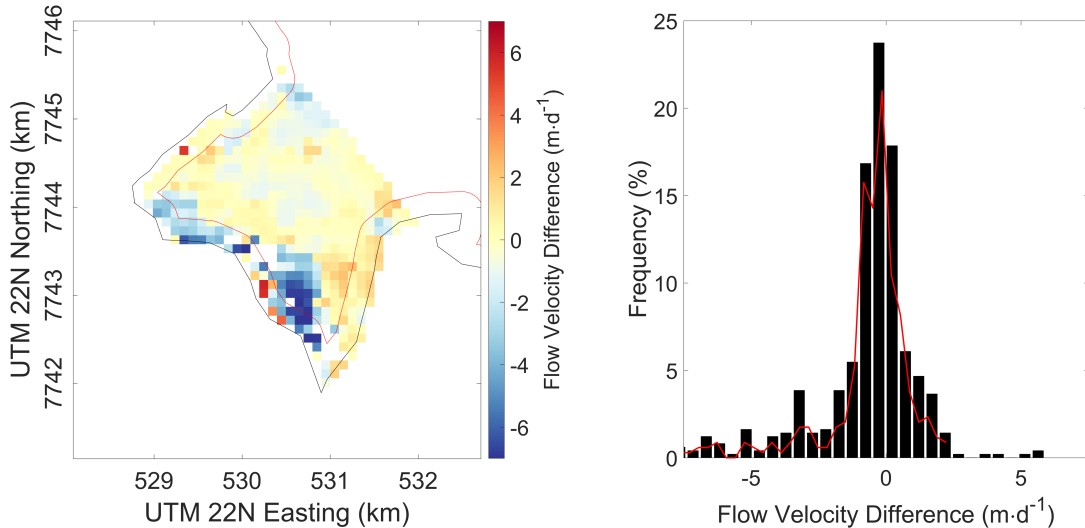

**Figure 7.** Distribution of flow velocity differences between S-1A satellite based and UAV based calculations. Negative values denote higher flow velocities from the UAV data set, positive values higher values based on the SAR data. The red line in the histogram shows the distribution of values when a 250 m buffer zone at the glacier's margins was excluded.

results substantially or is compensated enough by the chosen smaller patch sizes. The good representation of spatial velocity
gradients implies that strain rate fields are also robust which is crucial for constraining ice flow or calving models or process studies.

However, strong changes in backscatter occurring between two acquisitions (e.g. due to changes in temperature causing surface melt or precipitation) can cause a deterioration of the results, resulting in data voids after the outlier detection. This was especially the case for the acquisition of 2016/08/25 that was within a warm period with temperatures never falling below freezing for a week, while the corresponding acquisition on 2016/08/13 occurred during a period of pronounced diurnal variations of the temperature. Detection of small flow velocities (<2 md$^{-1}$) using our offset tracking from SAR intensity data can be impaired by low SNR, resulting in unreliable and noisy velocity estimates. An increase of the time between acquisitions for slower parts of the glacier, for example doubling the period to 24 days, can alleviate these issues, at the cost of reduced temporal resolution.

## 4.1 Influence of sample interval

To improve the representation of the velocity gradients even close to the glacier's terminus, a short sample interval followed by a downsampling step to e.g. $100{\times}100$ m is beneficial, although at a higher computational cost. When choosing too large a sampling step size (e.g. $40{\times}40$ px, i.e. $100{\times}100$ m), large gradients in flow velocity (such as areas close to the calving margin) might not be resolved, whereas a fine sample interval increases the chance of calculating offsets right up to the calving front without the reference window overlapping into the fjord area (cf. Fig. A9). Given this paper's focus on *validation* of

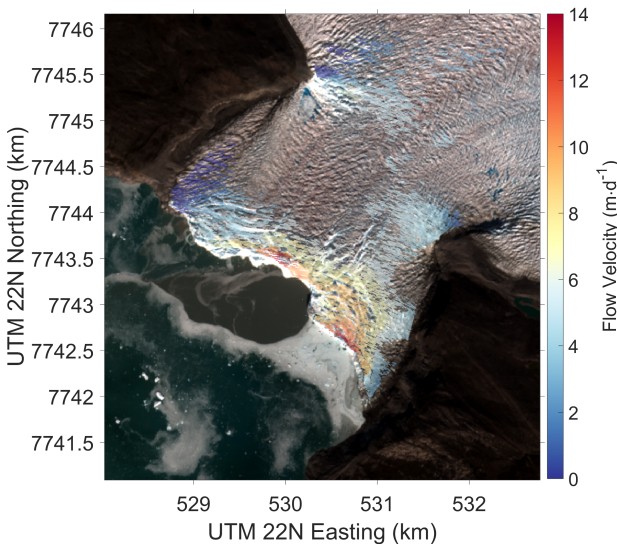

**Figure 8.** Flow speed map based on 4 day integrated TRI acquisitions between 2016/08/21 and 2016/08/25. The shown flow speeds represent the flow magnitude in line-of-sight direction towards the TRI sensor (cf. Fig. 1). No velocities could be derived for areas shadowed by the glacier's topography. Values with a coherence <0.6 or outside the glacier tongue were masked out. Clearly visible are the rifts close to the glacier's tongue, showing jumps in the glacier flow speed. Background: Sentinel-2A scene from 3 August 2016. Contains Copernicus Sentinel data (2016).

velocity estimates and spatial patterns using high-resolution reference data, the increased computational cost caused by a 2×2 pixel sample interval was acceptable. For processing at e.g. ice-sheet scale, choosing a coarser sample interval (e.g. 20×20 or 40×40 pixels) at the cost of some detail is advisable.

## 4.2 Uncertainties and sensitivities to patch size and acquisition period

As reported by Nagler et al. (2015) three main sources of error exist when using offset tracking for ice velocity estimation:

- – Errors in the matching procedure,

- – errors due to ionospheric disturbances,

- – geocoding errors.

Errors in the matching procedure are not only influenced by the image pair's co-registration and the quality of the amplitude
features, but also by the chosen size of the template (Nagler et al., 2015). This choice is not straightforward, as its optimality depends on the presence and prominence of features within. A *smaller* patch size can produce better results in regions with strong velocity gradients, yet suffer from increased noisiness. In contrast, a patch size that is too *large* might cause a blurring of the velocities, resulting in a trend towards lower values. This issue is investigated along a center profile in Fig. 10, corresponding

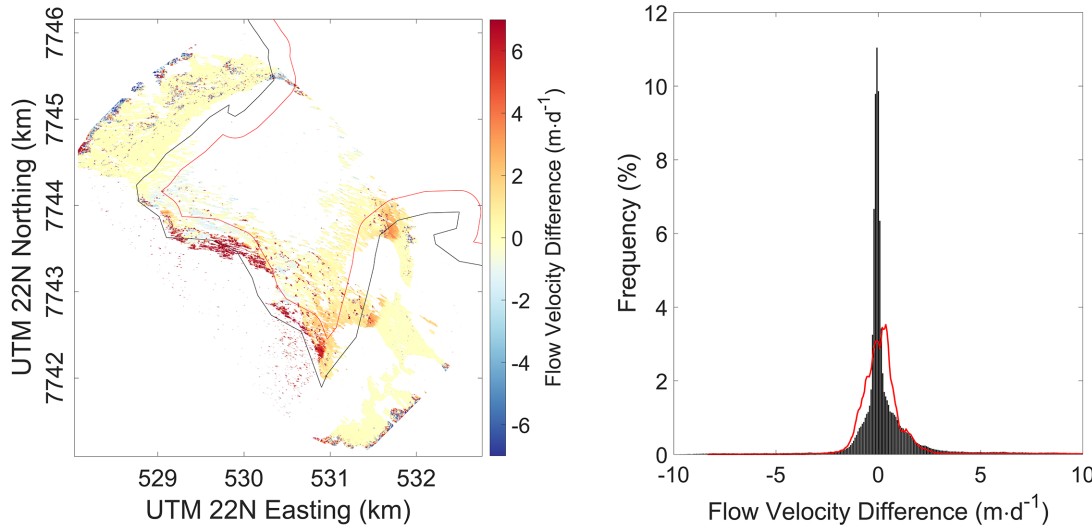

**Figure 9.** Distribution of flow velocity differences derived from UAV image matching and TRI interferometry. Positive values (red colors) depict higher estimated flow velocities from UAV, negative (blue colors) higher derived velocities for the interferometric TRI measurements. Only values inside the glacier's marginal boundary (black line) were included in the histogram. The red line in the histogram shows the distribution of values when a 250 m buffer zone at the glacier's margins was excluded. Note the velocity differences are from the line-of-sight components of the TRI.

to spatial extents of about 150×150 m, 250×250 m, and 350×350 m. A larger template size substantially reduces the noise,

but increases the area affected by edge effects such as the deceleration artifact at the calving front.

Furthermore, the temporal integration time of subsequent intensity trackings influences the results. While shorter integration times suffer from a higher noise level compared to products averaged over longer periods, they can be used to investigate short-term changes in flow dynamics of a glacier. This is illustrated in the sensitivity analysis of Fig. 11 showing SAR velocity results along the center flowline for different temporal averaging window sizes. Temporal averaging over several 12 day acquisitions

substantially smooths the data and in particular reduces the artefact of velocity reduction at the calving front. Regarding the geocoding process, i.e. the transformation from slant range to map projection, the errors introduced are primarily caused by DEM inaccuracies, as the geolocation accuracy of the S-1A/S-1B products has been shown to be well within the 7 m absolute location accuracy requirements specified by the European Space Agency (Schubert et al., 2017; Miranda et al., 2018; Piantanida et al., 2018). The DEM used for the project was the Greenland Ice Mapping Project (GIMP) DEM generated from data nominally from around 2007 at an original resolution of 30×30 m. We oversampled to 2.5×2.5 m. Since then, the slope and shape of glaciers has changed due to rapid thinning over the last decade. Joughin et al. (2018) report a horizontal location

5 error of ∼1.25 m for every 1 m of elevation error. Based on a comparison between the GIMP DEM and the TanDEM-X 90 m DEM (acquisition date 2012/06; Rizzoli et al., 2017), the surface lowering rates were 6 ma⁻¹ close to Eqip Sermia's calving

front and 2 ma$^{-1}$ at 17 km from the terminus, amounting to a maximum of ∼70 m of horizontal location error or less than one pixel in our product. Changes in slope and shape of the glacier need to be accounted for as well when comparing three-dimensional flow magnitudes or flow velocities assuming surface parallel flow, as they can introduce biases in the estimated magnitude of surface velocities (Nagler et al., 2015). Using the calculated lowering rates stated above, this results in a surface slope change of 0.15 degrees in 9 years. This value is almost identical to the one reported for Jakobshavn Glacier by Nagler et al. (2015), reporting a bias of 0.5% in the magnitude of surface velocity caused by surface lowering. Further, as *Eqip Sermia* is a fully grounded glacier, errors due to tidal influences can be neglected.

As we relied solely on the feature tracking methodology for the reasons explained in Sect. 2.2, errors are expected to be larger compared to those derived from interferometric-based approaches alone (Strozzi et al., 2002; Short and Gray, 2014). When comparing the derived mean error of 11.5% between our GPS measurements and the velocities derived from intensity tracking, our findings agree with the 9.7% difference reported by Ahlstrøm et al. (2013) using intensity tracking. Similar errors were found when comparing the SAR derived velocities to the high-resolution UAV data over the glacier tongue, with a mean and median difference of 12.4% and 8.5%, respectively. The differing spatial resolutions of the ground truth data sets used (i.e. UAV, TRI, GPS) resulted in additional uncertainties introduced during the resampling of the data to match the spaceborne acquisitions.

An analysis of the offsets in the easting and northing directions calculated over stable, non-moving terrain north and south of *Eqip Sermia* showed stable results with a mean velocity of <0.01 md$^{-1}$ and a standard deviation of <0.3 md$^{-1}$ in both the easting and northing directions over the 13 month period corresponding to the time span of the Greenland Ice Sheet CCI product (Nagler et al., 2015) (cf. Fig. 12).

### 4.3 Comparison to operational ice velocity products

Due to the large spatial coverage of currently available, operational ice flow velocity products such as products available from the Greenland Ice Sheet CCI (Nagler et al., 2015) and the National Snow & Ice Data Center's MEaSUREs Greenland Ice Sheet Velocity Map (Joughin et al., 2010, 2015, updated 2018), these products are only available for specific glaciers and for specific times at a high temporal resolution and do not cover *Eqip Sermia*. For our observation period, the monthly MEaSUREs Ice Velocity products are available at a spatial resolution of 200×200 m, while the Greenland-wide ice velocity map from the Greenland Ice Sheet CCI is only available on a yearly basis with a grid spacing of 500×500 m. In order to avoid effects from differing time periods, we compiled time-averages over the Greenland Ice Sheet CCIs 13 months time period (2015/10/01 - 2016/10/31) (Nagler et al., 2015) based on the velocities from all available 40 Sentinel-1 image pairs and 13 monthly MEa-SUREs ice velocity products (Joughin et al., 2015, updated 2018). Both operational products differ significantly from the flow velocities calculated in this study, both along the center flowline of the glacier (cf. Fig. 13) and along the two cross profiles (cf. Fig. 14 and 15). In addition to different processing methods, these differences are also likely a result of spatial smoothing in the operational products, and are most strongly pronounced towards the fast flowing frontal part of the glacier, where strong velocity gradients occur (cf. Fig. 13). There, and in contrast to our SAR- and UAV-derived velocities, the operational products indicate a slight or substantial deceleration in the vicinity of the terminus (cf. Fig. 13), likely an artefact from boundary effects

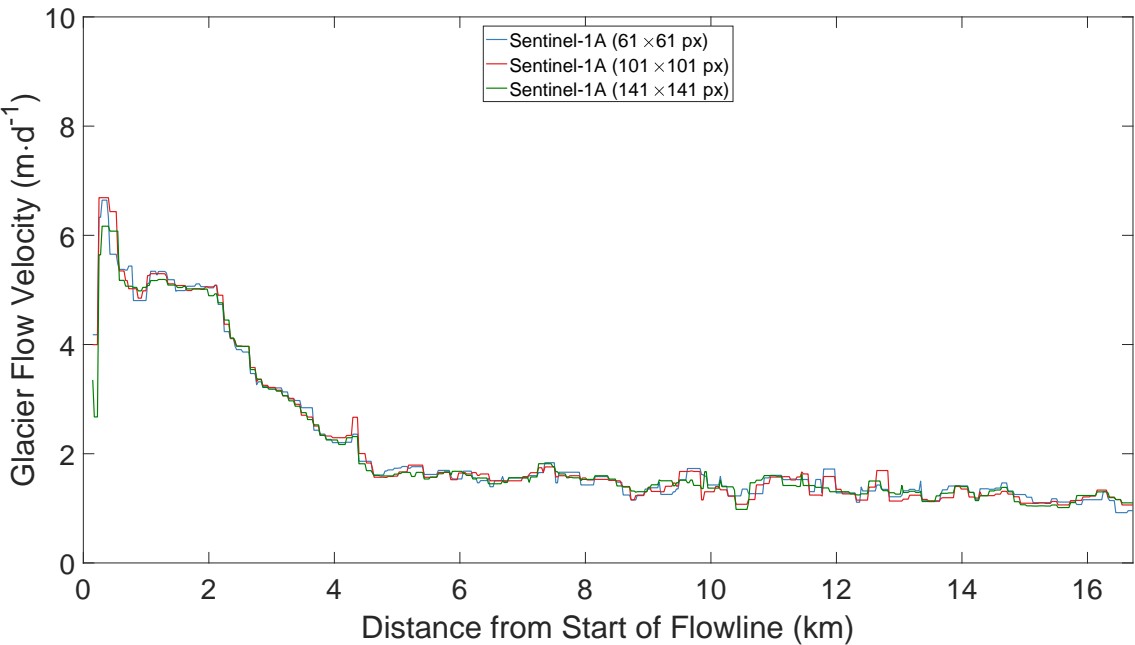

**Figure 10.** Mean flow velocity along the central flowline (cf. Fig. 1) between 2016/07/08 and 2016/09/20 for different template sizes. Smaller templates are better in capturing the velocity gradients occurring towards the glacier front, resulting in slightly higher flow velocities. Furthermore, the area affected by overlaps with surrounding areas of the glacier is diminished, resulting in reliable values closer to the glacier front. Bigger template sizes tend to result in smoother results.

and smoothing. The operational product appears to underestimate the velocities of the main tongue up to 3 km behind the front by about 10% to 20% as an effect that spans the whole width of the tongue (cf. Fig. 14). Our UAV-derived velocities
25  confirm this underestimation over the tongue and are even slightly higher than our SAR-results which may be an effect of the different acquisition period over 4 days in the early Arctic summer. Further upstream, the discrepancies between SAR and the operational products generally decreases, but near the centerline our SAR estimates at the location of the lowest GPS (EG09, Fig. 15) were still significantly higher. These findings are also valid when comparing the monthly MEaSUREs product from August 2016 with time-averages based on our Sentinel-1A image pairs (cf. Fig A2-A6). Despite the differences in flow veloci-
30  ties between our maps and the operational products, there is good agreement between the different products on direction along the flowline (calculated from the X- and Y-Offsets; cf. Fig. A8).

The above differences, calculated for the period between 2015/10/01 and 2016/10/31 (and similarly for the monthly MEa-SUREs product), imply that using the available, operational glacier flow velocity data sets for estimation of ice discharge with e.g. a flux-gate approach will result in an underestimation of ice flux (between 7% in comparison to MEaSUREs and 28% when compared to Greenland Ice Sheet CCI across the tongue, cf. Fig. 1), as such fluxes are calculated using surface velocity observations to approximate horizontal, depth-averaged ice velocity (Osmanoğlu et al., 2013). This underestimation cancels out if the focus is for example on changes in ice flux over time (Howat et al., 2011; Rignot et al., 2008). However, mass budget

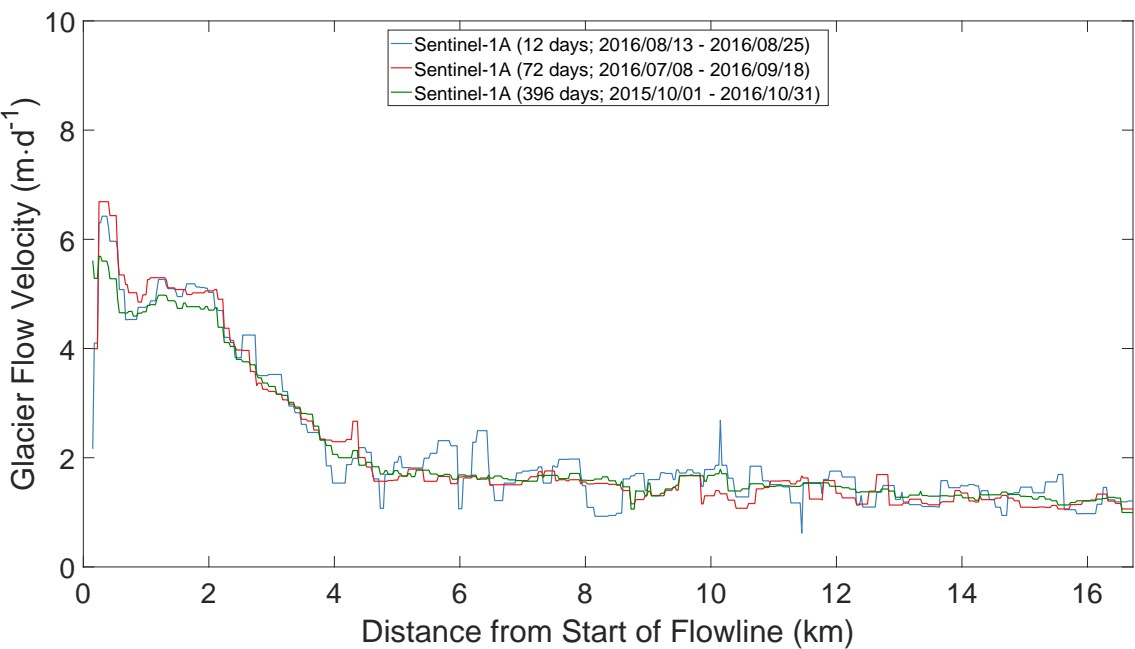

**Figure 11.** Mean flow velocity along the central flowline (cf. Fig. 1) for different temporal integration periods. Shorter periods are more prone to noisy results, but offer higher temporal resolution, while an increase in temporal integration time smoothens the results.

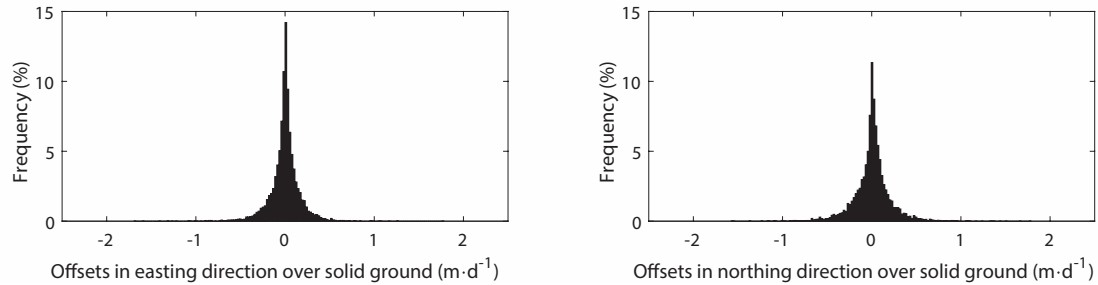

**Figure 12.** Distribution of velocity offsets in easting and northing direction over solid ground from SAR Intensity Tracking between 2015/10/01 and 2016/10/31 (13 months). Mean easting offset is -0.002 md$^{-1}$ with a standard deviation of 0.173 md$^{-1}$, mean northing offset is 0.003 md$^{-1}$ with a standard deviation of 0.214 md$^{-1}$.

methods take the difference between the absolute discharge and the surface mass balance integrated over the upstream catch-
ment and an underestimation in flow then systematically underestimates mass loss. Considering that the mass loss from the
mass budget calculation is only a fraction (a few 10%) of the total discharge at the terminus (Enderlin et al., 2014; Rignot and
Kanagaratnam, 2006), this 7% to 28% underestimation in near terminus ice flux would substantially affect mass loss estimates.
Of course this issue is less pronounced if the flux gates are located in the slower flowing parts upstream, but then an extra
estimation of mass changes downstream is still required. Given the parameter settings used to produce the operational products

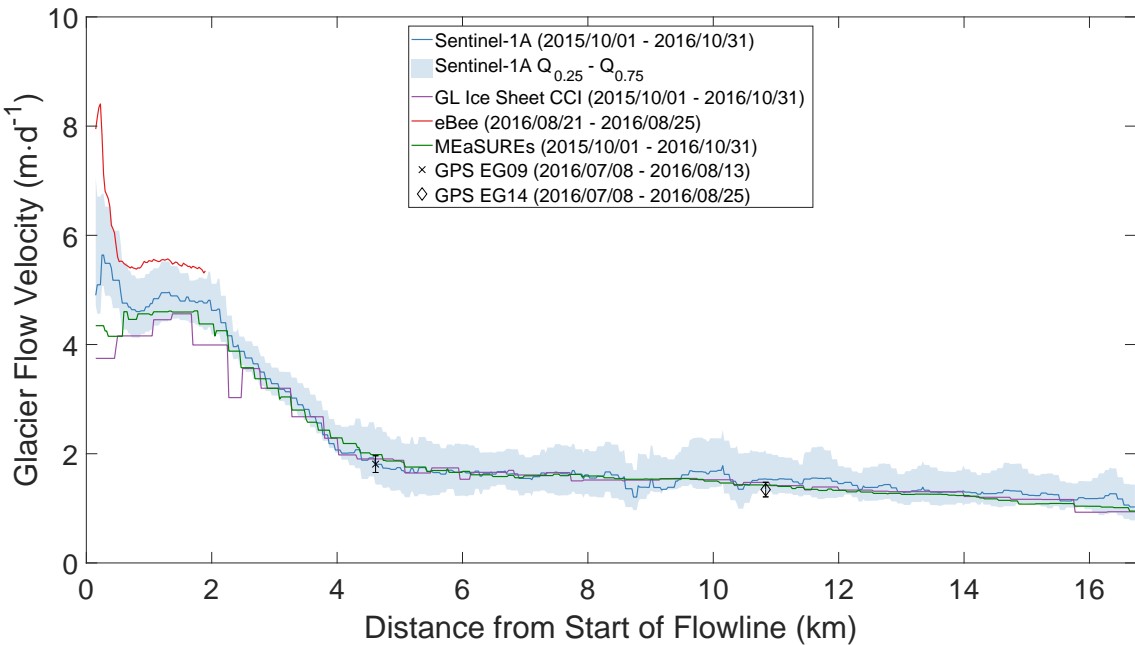

**Figure 13.** Mean annual flow speed along the central flowline (cf. Fig. 1) between 2015/10/01 and 2016/10/31 (13 months) for different products, starting at the glacier's terminus. In red, the reference flow speed based on the 4-day UAV mosaics acquired on 2016/08/21 and 2016/08/25 is shown.

(i.e. template window size, sampling step size, ground sampling distance), this underestimation in ice flow near the terminus, may likely apply also to other similar medium-sized outlet glaciers and hence have an impact on mass loss estimates of the whole Greenland Ice Sheet.

Our near-terminus flow-fields will also imply higher frontal strain rates (compared to the operational products) which affects observational constraints for models of flow dynamics of calving and our understanding of terminus dynamics (Nick et al., 15   2009, 2013; Choi et al., 2018; Morlighem et al., 2017; King et al., 2018).

## 5   Conclusions

The *in situ* measurements using a terrestrial radar interferometer (TRI), an unmanned aerial vehicle (UAV), and continuous GPS measurements used in this study were acquired during summer 2016 on a medium sized, ocean-terminating outlet glacier in West Greenland. This data was successfully used for validation of a standard SAR-based velocity offset tracking approach with 5   a specific focus on resolving the fast-flowing terminus area. We could show good agreement in magnitude and spatial pattern between multiple independent ground measurements and the flow velocities derived from different spaceborne C-band SAR sensors. The validity of the velocities derived using an iterative intensity tracking approach was demonstrated also for areas close to the glacier's calving front. Accurate near terminus velocity fields and related strain-rates are crucial for investigating

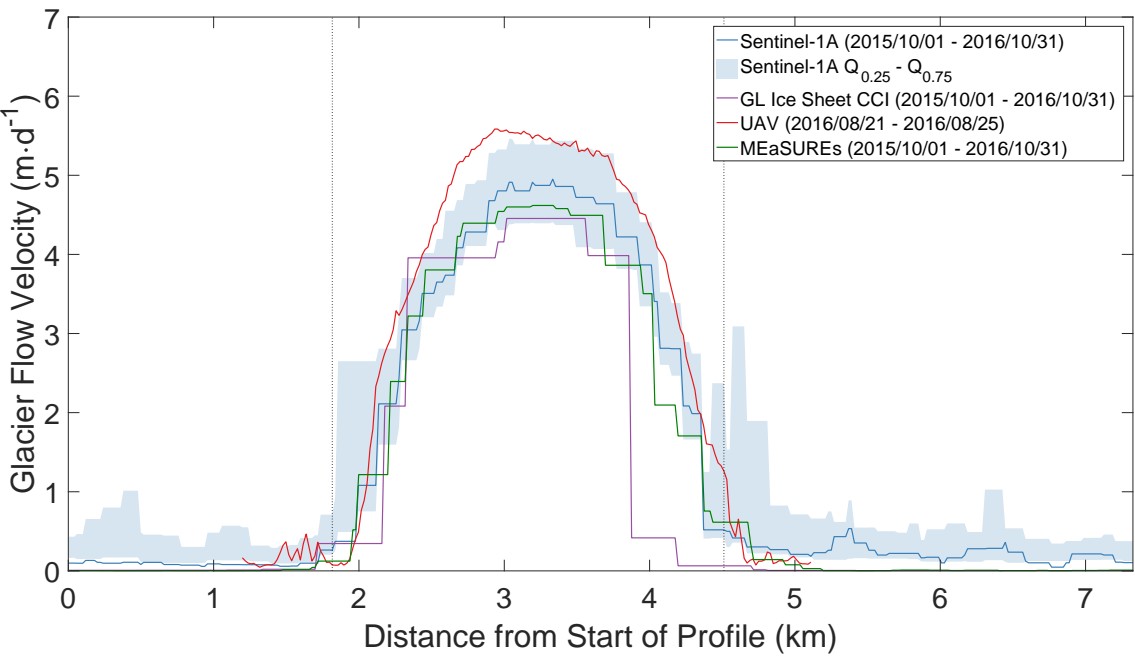

**Figure 14.** Mean annual flow speed across the glacier tongue (cf. Fig. 1) between 2015/10/01 and 2016/10/31 (13 months) for different products, starting at the orographic left side of the glacier. In red, reference flow speed based on the 4-day UAV mosaics acquired on 2016/08/21 and 2016/08/25 is shown. The dashed lines depict the glacier's margins.

the calving process, for constraining flow models and also ultimately for assessing ice flux and mass loss.

While there is a good agreement between the different data sets, caution should be exercised close to the glacier's margin, where the detection of its movements can be influenced by the chosen template size. While the size of this overlapping region can be minimized by choosing smaller template sizes, this introduces noise into the resulting velocity estimates due to erroneous correlations. Similarly, the upper boundary of the template size is limited by increased blurring, especially in regions where high velocity gradients occur. These limitations could be mitigated using an iterative matching approach with increasing template sizes. Filtering based on long-term directional information seemed successful in removing outliers, albeit at the cost of data voids, in particular towards the glacier margins.

Finally, we were able to demonstrate the feasibility of our offset tracking approach using spaceborne SAR intensity data to derive glacier flow velocities much closer to the calving front than standard operational products. Given the short orbital-repeat and illumination independency of Sentinel's SAR sensor, our approach has excellent potential for quasi-continuous operational derivation of accurate flow speed near calving termini and thus to provide velocity time-series for the analysis of terminus dynamics and ice sheet mass changes.

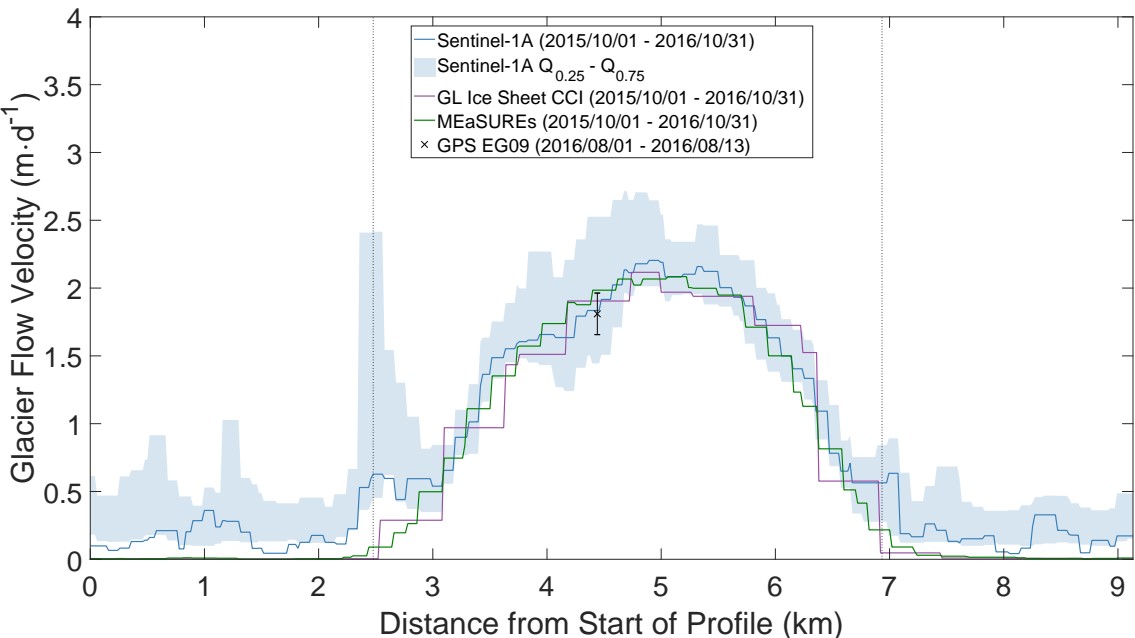

**Figure 15.** Mean annual flow speed across the position of GPS tracker EG09 (cf. Fig. 1) between 2015/10/01 and 2016/10/31 (13 months) for different products, starting at the orographic left side of the glacier. The dashed lines depict the glacier's margins. Note: the peak at the orographic left margin emerges from a debris covered marginal moraine.

*Data availability.* Data from this study can be made available from the authors upon request. The Sentinel-1 SAR data are available through the ESA Copernicus Science Hub: https://scihub.copernicus.eu/.

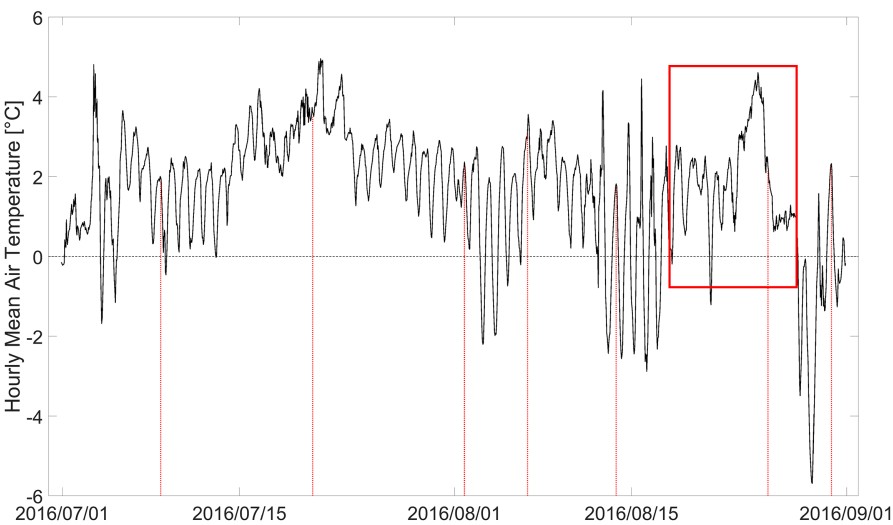

**Figure A1.** Hourly Mean Temperatures in degrees Celsius for GCNet Station JAR1 (69°29'42" N, 49°42'14" W, 932 m a.s.l.; Steffen et al., 1996) in black, the satellite acquisition times shown in red. Highlighted by the red rectangle is the warm period starting on August 21[st] until August 28[th].

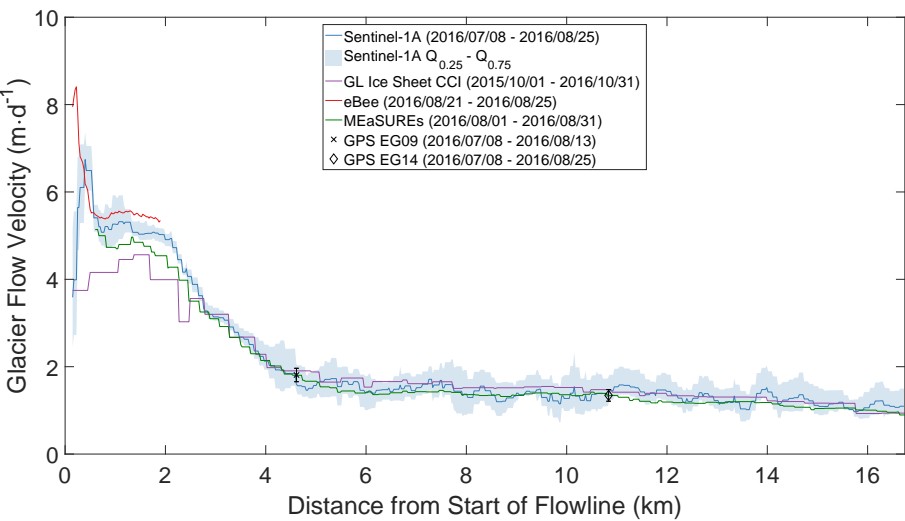

**Figure A2.** Mean flow speed along the central flowline (cf. Fig. 1) derived from S1A scenes acquired between 2016/07/08 and 2016/08/25 (48 days), starting at the glacier's terminus. In red, reference flow speed based on the 4-day UAV mosaics acquired on 2016/08/21 and 2016/08/25, in green and purple flow velocities from MEaSUREs and Greenland Ice Sheet CCI products are shown.

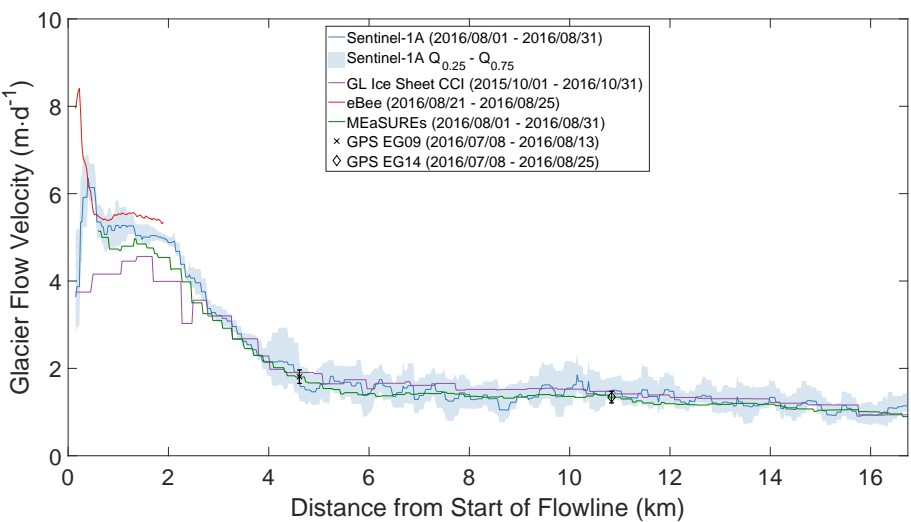

**Figure A3.** Mean flow speed along the central flowline (cf. Fig. 1) derived from S1A scenes acquired between 2016/08/01 and 2016/08/31 (1 month), starting at the glacier's terminus. In red, reference flow speed based on the 4-day UAV mosaics acquired on 2016/08/21 and 2016/08/25, in green and purple flow velocities from MEaSUREs and Greenland Ice Sheet CCI products are shown.

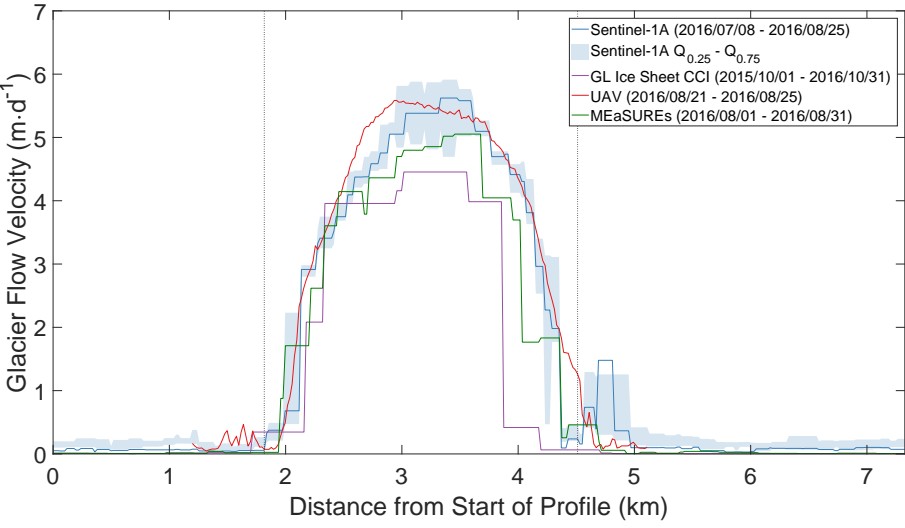

**Figure A4.** Mean flow speed across the glacier tongue (cf. Fig. 1) derived from S1A scenes acquired between 2016/07/08 and 2016/08/25 (48 days), starting at the orographic left side of the glacier. In red, reference flow speed based on the 4-day UAV mosaics acquired on 2016/08/21 and 2016/08/25, in green and purple flow velocities from MEaSUREs and Greenland Ice Sheet CCI products are shown. The dashed lines depict the glacier's margins.

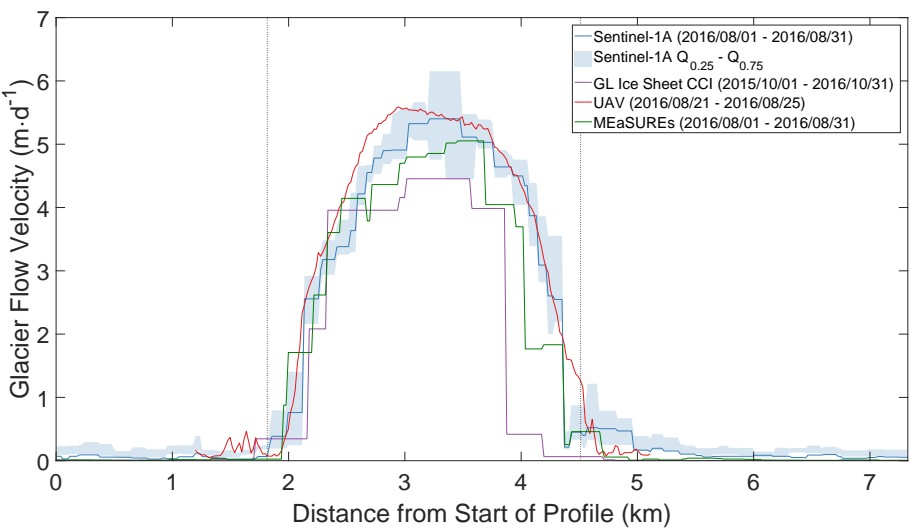

**Figure A5.** Mean flow speed across the glacier tongue (cf. Fig. 1) derived from S1A scenes acquired between 2016/08/01 and 2016/08/31 (1 month), starting at the orographic left side of the glacier. In red, reference flow speed based on the 4-day UAV mosaics acquired on 2016/08/21 and 2016/08/25, in green and purple flow velocities from MEaSUREs and Greenland Ice Sheet CCI products are shown. The dashed lines depict the glacier's margins.

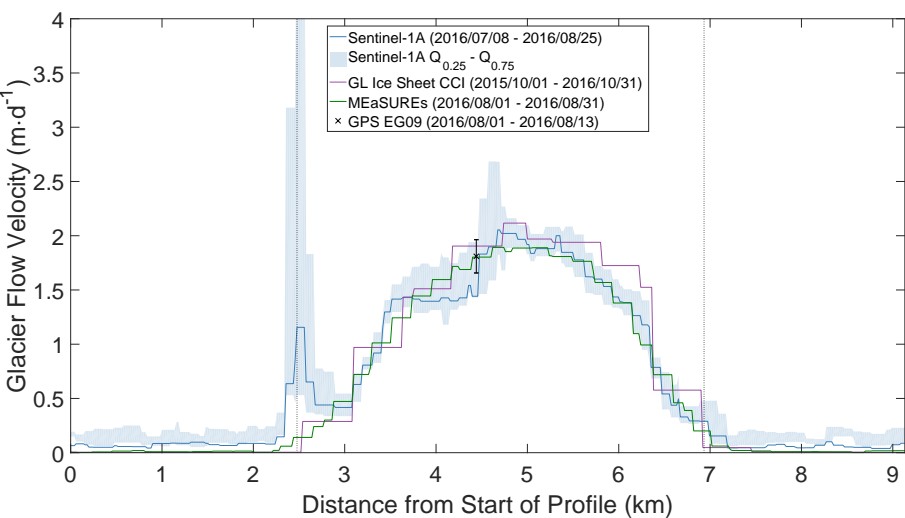

**Figure A6.** Mean flow speed across the position of GPS tracker EG09 (cf. Fig. 1) derived from S1A scenes acquired between 2016/07/08 and 2016/08/25 (48 days), starting at the orographic left side of the glacier. In green and purple reference flow velocities from MEaSUREs and Greenland Ice Sheet CCI products are shown. The dashed lines depict the glacier's margins. Note: the peak at the orographic left margin emerges from a debris covered marginal moraine.

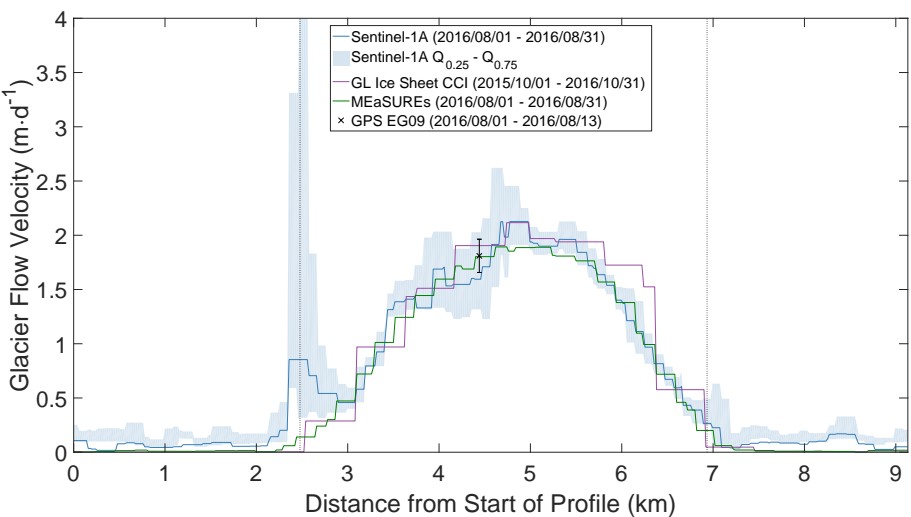

**Figure A7.** Mean flow speed across the position of GPS tracker EG09 (cf. Fig. 1) derived from S1A scenes acquired between 2016/08/01 and 2016/08/31 (1 month), starting at the orographic left side of the glacier. In green and purple reference flow velocities from MEaSUREs and Greenland Ice Sheet CCI products are shown. The dashed lines depict the glacier's margins. Note: the peak at the orographic left margin emerges from a debris covered marginal moraine.

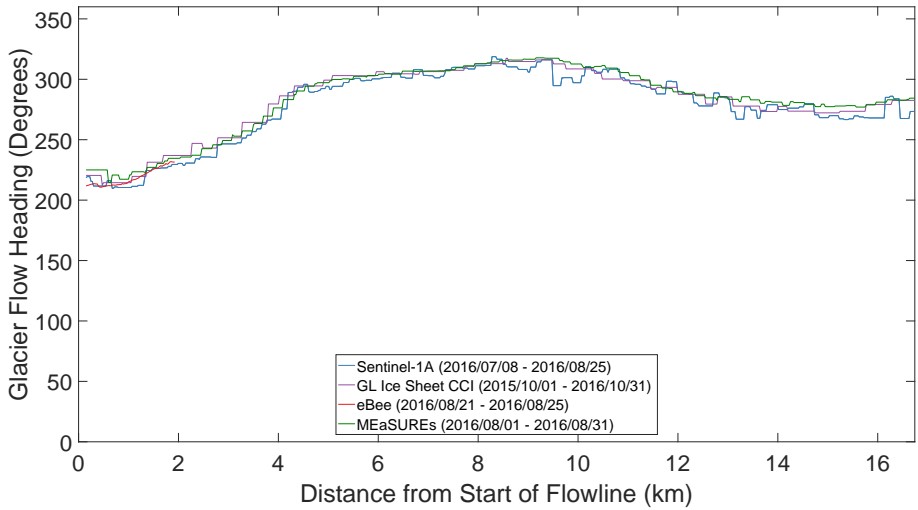

**Figure A8.** Mean flow direction in degrees referenced to north along the central flowline (cf. Fig. 1) derived from S1A scenes acquired between 2016/07/08 and 2016/08/25 (48 days), starting at the glacier's terminus. In red, reference flow heading based on the 4-day UAV mosaics acquired on 2016/08/21 and 2016/08/25, in green and purple flow direction from MEaSUREs and Greenland Ice Sheet CCI products are shown. Mean absolute differences along the flowline are 5.8° between our SAR-based results and the Greenland Ice Sheet CCI product, and 6.1° when compared to the product from MEaSUREs.

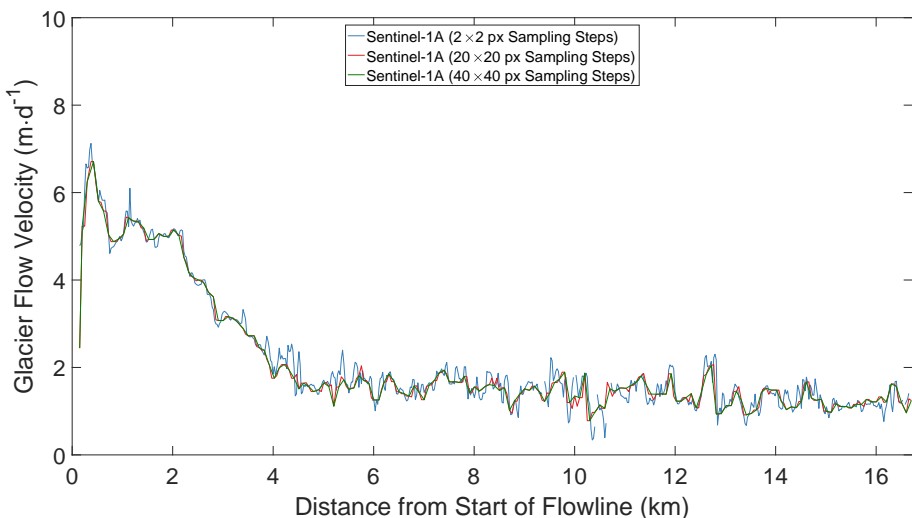

**Figure A9.** Mean flow speed along the central flowline (cf. Fig. 1) for different sampling step sizes derived from S1A scenes acquired between 2016/07/08 and 2016/08/25 (48 days), starting at the glacier's terminus.

*Author contributions.* C. Rohner, D. Small, D. Henke, and A. Vieli designed the manuscript. C. Rohner generated the flow fields, processed the UAV data and and performed all analysis. D. Small prepared the processing of the Sentinel-1 and RADARSAT-2 data. M. P. Lüthi performed TRI measurements and processing. C. Rohner wrote the draft of the manuscript. All authors contributed to the final version of the manuscript.

*Competing interests.* The authors declare that they have no conflict of interest.

*Acknowledgements.* This work was funded by the Swiss National Science Foundation Grant 200021_156098. RADARSAT-2 scenes were provided through the MDA/CSA SOAR Programme (Project #16821). We thank the European space agency (ESA) for providing the Sentinel-1 data. Lastly, we thank the reviewers for their valuable work, which led to significant improvements in the revised manuscript.

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
