# Peer review of "Multisensor validation of tidewater glacier flow fields derived from SAR intensity tracking"

_The Cryosphere, 2018_

## Referee Comment (RC1) · Anonymous Referee #1 · 20 Feb 2019

General comments

Rohner et al. present a detailed validation study of ice velocity measurements derived from intensity feature tracking using Sentinel-1A and RADARSAT-2 SAR images. The study area is a tide-water glacier in Western Greenland (Eqip Sermia) which showed substantial fluctuations in recent years. The derived velocity maps are compared with field data, derived from 3 GPS stations, a Terrestrial Radar Interferometer (TRI) and repeated UAV surveys, in an effort to demonstrate the accuracy and limits of the method and data sets. Statistics are provided of the outcomes and are visualised in histograms, scatter plots and difference maps. The authors report a good agreement with the in-situ data with minor differences near the calving front ascribed to the characteristics of the different techniques. The ice velocity maps are also compared with Greenland-wide

products, produced as part of the NASA MEaSUREs and ESA CCI programs, revealing significant differences. According to the authors these products underestimate ice flow, resulting in an underestimation of ice discharge and an overestimation of mass loss when using the products for mass balance assessments.

The subject of this paper, the assessment of inherent uncertainties in satellite data products with field derived data, is a topic of great relevance for quantifying the quality of these products. The combination of different contemporaneously acquired and independent (field) measurement techniques, as described in this study, provides hereby the highest level of validation. The paper also exemplifies the added value for ice velocity maps with high spatial and temporal resolution. From these perspectives, this study is a relevant and welcome contribution to the existing literature and within the scope of The Cryosphere. However, there appears to be a serious flaw in the core data set, i.e. the Sentinel-1 and RADARSAT-2 derived ice velocity maps, which necessitates major revisions and re-analysis of the data. To clarify, the authors claim to have produced ice velocity maps at a spatial resolution of 5x5m. This, however, is not possible with the SAR data sets that they have used (or any spaceborne sensor for that matter). At best, one can produce a velocity map with a 5m grid spacing (which is not the same as resolution), but the required step-size would lead to a ∼99% overlap of image patches, making them strongly correlated and they can therefore not be treated as independent measurements. With Sentinel-1 SAR in IW mode, it is likely not possible to achieve a much higher resolution than 100m using intensity feature tracking. Unfortunately, this issue renders most of the subsequent analysis futile and precludes drawing any major conclusions from the study. There are also several other issues and logical fallacies in the manuscript which I detail below.

Specific comments

* As mentioned, the authors claim to have produced Sentinel-1 and RADARSAT-2 derived velocity maps with, in their words, a spatial resolution of 5m (see Pg. 5 & Pg. 14, Ln 13). This is surprisingly at a similar or even higher resolution than the satellite sensor. As mentioned above, with the selected patch sizes (250mx250m) it does not make much sense to produce maps with a spacing lower than 100m, the resolution is not increased but you basically end up with a smoothed dataset with no extra information and not suitable for for example modelling purposes.

* Pg. 5 Ln 11-12: "for each of the 133 image pairs available between 2014/10/11 and 2018/03/18". In fact, many more image pairs are available when considering tracks in both ascending and descending direction.

* Section 2.1 & 2.2: The data and methods section seem to miss some essential information required for a careful interpretation of the study results. For example, how was the coregistration performed? Were any ground control data used or only orbit data? Was the SLC data deramped for the azimuth phase ramp? How and when is the burst and sub-swath stitching performed? It appears that offset tracking was performed on terrain corrected geocoded images. Why not in SAR geometry as errors in the DEM are less of a concern and this approach would cause the least distortion as no resampling is required. Also, it is not clear if the SLC data was oversampled before converting to amplitude. Which components of velocity are provided in the output, what assumptions are made (horizontal, vertical, slope parallel?). How is dealt with radar shadow, which could be a concern in steep terrain, in particular when using only ascending data as done here?

* Section 3: Much effort is spent on intercomparing the acquired field data with the satellite derived velocity maps but it seems to focus only on comapring velocity magnitude. Because the accuracy might differ in different directions, it would be useful to do an intercomparison component wise. Also, a concluding section on the final error estimate of the ice velocity maps, integrating the outcome of all the independent estimates, is missing.

* Pg. 9 Ln 8: "which corresponds to an area of about 25×25 m." Considering the chosen template size used for the tracking it actually corresponds to a much larger

area.

* Pg. 16 Ln 10: ". Left uncorrected, these introduce biases in the estimated magnitude of surface velocities (Nagler et al., 2015). ". Did the authors apply any such corrections? If not, what is the estimated bias introduced by this?

* Section 4.2: A large part of the discussion concerns an intercomparison of an annually averaged version of the '5m' maps with products from MEaSUREs and CCI. The authors find substantial differences near the calving front and margins but also further upstream. These differences are reported as an underestimation of ice flow in the operational products and according to the authors this implies an underestimation of ice fluxes and an overestimation of Greenland mass loss when used for mass balance assessments. These claims are, however, unsubstantiated by the current work for a number of reasons, aside from the issue regarding resolution described before.

Firstly, the reported differences might as well imply an overestimation of the annual product. How is this distinguished? Although the authors mention their claim is supported by the intercomparisons (GPS, UAV, TRI), these only involve short term intercomparisons during a number of episodes in summer when ice flow is usually faster than the annual mean (Pg. 16 ln 17). Also, looking in closer detail at the GPS intercomparison (section 3.2), there appears to be a systematic underestimation for most of the data points (Figure 5). In particular with Sentinel-1 there are large differences, that seem much higher than the reported 8.7% and are up to nearly 40% (Table 3). In contrast, NASA MEaSUREs reports much better agreement with in-situ GPS (Joughin et al., 2017, 2018).

Secondly, even if for this medium-sized outlet glacier ice flow is underestimated in standardized products, this cannot be generalized into a systematic and substantial underestimation of Greenland ice fluxes, as the authors assert (Pg. 17 Ln 21-30). The nuance seems to be missing here and it appears the paper is overreaching while downplaying uncertainties.

Thirdly an underestimation of flow velocities on an outlet glacier would indeed lead to an underestimation of ice fluxes, but this would also lead to an underestimation of mass loss in the mass budget calculation as less ice is exported. It is unclear why or how this would lead to an overestimation of mass loss as stated by the authors.

In order to clarify the discrepancies addressed above, it is necessary to better explain the methods and assumptions used, and to check and revise the error estimates. It could be that perhaps different components are compared (e.g. surface parallel velocity vs horizontal velocity). Also, perhaps the authors are not aware that 250m CCI products are available (Nagler et al., 2015).

\* In several places throughout the manuscript the authors claim their velocity maps as 'improved' over existing products. I have no doubt that these products can be improved in several meaningful ways, including by increasing the spatial/temporal resolution or for example by correction of ionospheric streaks. But, aside from the 'increased' spatial resolution there does not seem to be any further methodological improvement to warrant this claim. Concerning temporal resolution, the CCI project has also provided time series at high temporal resolution, with temporal sampling up to every 6 days, albeit only for selected glaciers (see: http://esa-icesheets-greenland-cci.org). The presented study provides only 12-day maps.

Figures

Figs 4, 6 & 8: Use same colour scale. The high density of the flow vectors in fig 4 & 6 obscures the velocity map.

Fig 9: Most of the data points seem to lie outside of the glacier (on bedrock?), is this data included in the in the intercomparison?

Figs 10-15: Difficult to distinguish between the red and orange lines, I would suggest using green instead.

Fig 14 & 15: The glacier margins where velocity goes down to zero are missing in this

[Figure]

plot. These shear margins are good areas to show the improvement of the increased resolution.

Figs 14 & 15: The x-axis label mentions 'Distance from start of flowline', I assume 'start of profile' is mentioned as this concerns a cross profile.

References

Joughin, I., Smith, B. E., and Howat, I.: Greenland Ice Mapping Project: ice flow velocity variation at sub-monthly to decadal timescales, The Cryosphere, 12, 2211-2227, https://doi.org/10.5194/tc-12-2211-2018, 2018.

Joughin, I., B. Smith, and I. Howat. 2017. A complete map of Greenland ice velocity derived from satellite data collected over 20 years, Journal of Glaciology. 1-11. http://dx.doi.org/10.1017/jog.2017.73

Nagler, T.; Rott, H.; Hetzenecker, M.; Wuite, J.; Potin, P. The Sentinel-1 Mission: New Opportunities for Ice Sheet Observations. Remote Sens. 2015, 7, 9371-9389.

---

## Referee Comment (RC2) · Anonymous Referee #2 · 12 Apr 2019

General comments

The present study presents ice velocity estimates derived from an intensity tracking algorithm using Sentinel-1a and Radarsat-2 SAR images compared to independent measurements at Eqip Sermia. The independent dataset comes from in situ measurements from differential GPS, Terrestrial Radar Interferometer (TRI), UAV surveys, and from operational ice velocity products, NASA MEaSURES and ESA CCI. The study reports good agreement when compared to the field data, and shows 10%-20% difference compared to operational ice velocity products. The comparison of ice velocity derived from satellite missions and in situ datasets is extremely important, it makes a useful contribution to the field, and presents interesting results. However, the processing chain used to derive ice velocity from the satellite images, as pointed by Reviewer1,

lacks detailed description, and then affects the majority of the results in this paper.

For these reasons, and the specific comments listed below, I recommend the manuscript requires Major Revision before publication.

Specific comments:

P1L21-23: Any reference?

P2L7-8: A lot of efforts have been made to derive velocity with optical instruments.

P2L13-15: Add some more details about the Sentinel-1 mission, or at least some more references.

P3L7: "high-resolution": Spatial? Temporal? Both?

P3L14-17: It is hard to follow. Rewrite the sentence please.

P3L29: "RS-2" was not defined before.

P4L5: Which version of GIMP do you use? 30m, 90m? How does the DEM oversampling affect the uncertainty in the velocity products?

P5L19-21: Reword this sentence. Looks like meteorological conditions does not affect the velocity estimation, which is not true.

P5L6-7: As pointed by Reviewer1, how does your velocity results have resolution of 5x5m if S1 spatial resolution is 5x20m?

P5L9-12: From where, and how long is this "long-term flow velocity average product" ?

P7L10-11: Please give a brief description of the three-step approach.

P9Figure4: It's hard to see the colour range on the map behind the arrows. The arrows are great, however there is no scale and they are too small. Please add a scale bar, just to make it easier to the reader.

P10L8: The date 2016/08/13. Typo?

P11Figure6: Same as Figure 4. It's hard to see the colour range on the map behind the arrows. The arrows are great, however there is no scale and they are too small. Please add a scale bar, just to make it easier to the reader.

P12Figure7a: I'd include the Sentinel 2 image in the background as a reference for the masked areas.

P14Figure9a: Again, it's very hard to picture the excluded areas.

P16L9-10: Again, what is the spatial resolution of the DEM?

P16L3-5: "Due to the large spatial coverage ...". I suggest the authors to rewrite this sentence. NASA MEaSURES and ESA CCI are the two main operational centres that deliver velocity products. If the authors look carefully, they do provide high temporal velocity products to specific glaciers.

P17L21-24: I suggest the authors to rewrite the sentence. It presents a strong argument, taking in account that this work only provides measurements of a couple of seasons.

P20: "Finally, we were able to demonstrate...": Other studies have already demonstrated the ability of Sentinel-1 to estimate ice velocity near the ice front (e.g. Nagler et al., 2015; Joughin et al, 2018; Lemos et al., 2018). These studies also demonstrated improved results using shorter temporal baseline provided by Sentinel-1 (6 days) since Oct/2016, and the potential of Sentinel-1 to extend existent ice velocity time-series.

References:

Joughin, I.; Smith, B. E.; Howat, I., 2018: Greenland Ice Mapping Project: Ice Flow Velocity Variation at sub-monthly to decadal time scales. The Cryosphere., 12, 2211–2227.

Lemos, A.; Shepherd, A.; Mcmillan, M.; Hogg, A. E.; Hatton, E.; Joughin, I., 2018: Ice velocity of Jakobshavn Isbræ, Petermann Glacier, Nioghalvfjerdsfjorden and Zachariæ

Isstrøm, 2015-2017, from Sentinel 1-a/b SAR imagery. The Cryosphere., 12, 2087–2097.

Nagler, T.; Rott, H.; Hetzenecker, M.; Wuite, J.; Potin, P., 2015: The Sentinel-1 Mission: New Opportunities for Ice Sheet Observations. Remote Sensing., 7, 9371–9389.

---

## Author Comment (AC1) · 31 May 2019

Please see supplement to this comment including the following documents:

– The response to the reviewers (Response_ReviewerComments.pdf)

– Difference map GL Ice Sheet CCI Flow Velocities/ENVEO Cryoportal Flow Velocities (Map_Difference_CCI_ENVEOCryoportal.png)

– The revised manuscript with track changes (RevisedManuscript_LatexDiff.pdf)

Please also note the supplement to this comment:
https://www.the-cryosphere-discuss.net/tc-2018-278/tc-2018-278-AC1-supplement.zip

---

## Author Response (AR1)

We would like to thank the reviewers for their positive feedback regarding the importance and relevance with regard to validation and terminus area of our paper and their valuable comments on this manuscript.

Below, all points raised (in italics) are addressed (with responses directly below) and changes to the text are presented after each point in bold fonts. Additions to the tracked changes file are written in blue, omissions in red.

In brief, the main changes in the revised manuscript include:

- We resampled the velocity data to 100x100 m in order to avoid issues due to spatial correlation, but still resolve velocity gradients towards the front and at the margins,
- Accordingly, we recalculated all the statistics and replotted all the figures,
- We added an additional comparison and analysis of short-term (summer months) results to other products (monthly MEaSUREs product) for cross and along-flow profiles,
- We added some clarifications and details to the methods,
- We differentiated the discussion regarding impact on mass loss estimates and degree wo which the study is representative,
- We addressed all the minor editing issues.

**REFEREE COMMENTS #1**

General Comments

First, we respond to the main points raised by reviewer #1:

| 1.1 | *"The authors claim to have produced ice velocity maps at a spatial resolution of 5x5 m. This, however, is not possible with the SAR data sets that they have used (or any spaceborne sensor for that matter). At best, one can produce a velocity map with a 5m grid spacing (which is not the same as resolution), but the required step-size would lead to a ~99% overlap of image patches, making them strongly correlated and they can therefore not be treated as independent measurements. With Sentinel-1 SAR in IW mode, it is likely not possible to achieve a much higher resolution than 100m using intensity feature tracking."* |
|---|---|
| | We fully agree that a spatial resolution of 5x5 m is no attainable given the S1 SAR data sets used, but that the 5 m rather refers to the sampling interval (Pg. 5, Ln 21). As pointed out correctly by Reviewer 1, we unintentionally did not consequently use this term, but wrongly referred to it as "resolution". This has been changed throughout the document.
Our intention was to produce flow velocities at a high temporal resolution and a small sampling step size in order to get as close as possible to the glacier's calving terminus. When choosing a larger sampling step size (e.g. 250 m), some gradients in flow velocity (such as areas close to the calving margin) might not be resolved, whereas a small sampling step size increases the chance of calculating offsets right up to the calving front without the reference window overlapping into the fjord area. An additional figure showing the influence of different sampling step sizes was added to the appendix (cf. Fig. A9) and indicated in the text. In addition, we added a short discussion on the effect of the sample interval on flow velocity retrieval (Sect. 4.1).
To account for the correlated data points, we downsampled (by using the median) all offset data sets to a ground sampling distance of 100x100 m. We recalculated all figures and statistics and adjusted the corresponding values and text passages accordingly. It |

| | should be noted that in general, the findings and conclusion did not substantially change by this downsampling. The change in methodology is reflected in Sect. 2.2.1. |
|---|---|
| | **Addition to text: "To improve the representation of the velocity gradients even close to the glacier's terminus, a short sample interval followed by a downsampling step to e.g. 100×100 m is beneficial, although at a higher computational cost. When choosing too large a sampling step size (e.g. 40×40 px, i.e. 100×100 m), large gradients in flow velocity (such as areas close to the calving margin) might not be resolved, whereas a fine sample interval increases the chance of calculating offsets right up to the calving front without the reference window overlapping into the fjord area (cf. Fig. A9)."**

**Addition to text: "Following the outlier detection and process iteration step, $V_{map}$ was downscaled to a pixel spacing of 100×100 m by applying a median filter to account for the spatial correlation of adjacent pixels caused by overlapping template patches in the initial $V_{map}$ (cf. Sect. 2.2)"** |

Specific Comments

| 1.2 | *"As mentioned, the authors claim to have produced Sentinel-1 and RADARSAT-2 derived velocity maps with, in their words, a spatial resolution of 5m (see Pg. 5 & Pg. 14, Ln 13). This is surprisingly at a similar or even higher resolution than the satellite sensor. As mentioned above, with the selected patch sizes (250mx250m) it does not make much sense to produce maps with a spacing lower than 100m, the resolution is not increased but you basically end up with a smoothed dataset with no extra information and not suitable for example modelling purposes."* |
|---|---|
| | This was addressed in the answer to comment 1.1 |

| 1.3 | *"For each of the 133 image pairs available between 2014/10/11 and 2018/03/18". In fact, many more image pairs are available when considering tracks in both ascending and descending direction."* |
|---|---|
| | Due to an error, the number of image pairs had been calculated for the time until 2017/03/18, not until 2018/03/18 as had been stated in the manuscript. The correct number of image pairs, with less than 25 days difference, is a total of 256, which has now been adjusted accordingly throughout the text. |

| 1.4 | *"Section 2.1 & 2.2: The data and methods section seem to miss some essential information required for a careful interpretation of the study results. For example:*
    1. *how was the coregistration performed?*
    2. *Were any ground control data used or only orbit data?*
    3. *Was the SLC data deramped for the azimuth phase ramp?*
    4. *How and when is the burst and sub-swath stitching performed?*
    5. *It appears that offset tracking was performed on terrain corrected geocoded images. Why not in SAR geometry as errors in the DEM are less of a concern and this approach would cause the least distortion as no resampling is required?*
    6. *Also, it is not clear if the SLC data was oversampled before converting to amplitude?*
    7. *Which components of velocity are provided in the output?*
    8. *what assumptions are made (horizontal, vertical, slope parallel)?*
    9. *How is dealt with radar shadow, which could be a concern in steep terrain, in particular when using only ascending data as done here?"* |
|---|---|

To clarify the terrain correction and geocoding process, we adjusted the text and included further references (see below).

1. As we geocoded the individual scenes before performing intensity tracking, we did not need to co-register the slant-range images
2. Only orbit data were used, no ground control data
3. Deramping was not necessary, as debursting and detection were performed before geocoding (amplitudes only)
4. Debursting, mosaicking, detection and multi-looking are all performed (in that order) by in-house software, based on the S-1 SLC input products obtained from the Copernicus SciHub.
5. We opted to use geocoded images as this results in a constant ground sample spacing for the whole scene. As for resampling uncertainties: coregistration also results in uncertainties which are carried over to the final result. Additionally, geocoding the vector field (to create a vector field in map geometry) would create resampling uncertainties as well. In short, some uncertainties are unavoidable, and we prefer to perform tracking in map geometry for practical reasons. As long as the temporal and spatial variations of the real signal are great enough, we do not believe the order of the processing steps plays a significant role.
6. The SLC data was not oversampled. The SLC samples were detected and multi-looked prior to the geocoding step.
7. Our intensity tracking methodology provides pixel offsets in X- and Y- direction, corresponding to offsets in Easting and Northing given the UTM 22N coordinate system of the input scenes
8. The resulting velocities represent horizontal displacements. Potential DEM effects have not been taken into account; however, this facilitates comparability of the estimates with products from other sources and over longer time series.
9. Radar shadow was not a concern in our case as the glacier's geometry and the surrounding topography did not cause big areas with radar shadow. Further, the shadows cast in small crevasses help to increase the texture contrast and hence the correlation.

**Changes in text: "The detected HH polarized SAR images from both sensors were geometrically terrain corrected using Range-Doppler geocoding (Meier et al., 1993) based on the "GIMP" Digital Elevation Model (DEM). The Greenland Ice Mapping Project (GIMP) DEM has a grid spacing of 30×30 m (Version 2.1; Howat et al., 2014), which was oversampled to 2.5×2.5 m. As we operated in the DEM geometry, no separate co-registration was performed. No tiepoints were employed during geocoding, as the geolocation accuracy was sufficient (Schubert et al., 2017)."**

| 1.5 | *"Section 3: Much effort is spent on intercomparing the acquired field data with the satellite derived velocity maps, but it seems to focus only on comparing velocity magnitude. Because the accuracy might differ in different directions, it would be useful to do an intercomparison component wise."* |
|---|---|
| | The intercomparison between our results with MEaSUREs, CCI, and UAV showed a good agreement regarding the flow direction based on the X- and Y-Offsets, with an average absolute difference of ~6° when compared to MEaSUREs and CCI products. An additional diagram including flow direction of the different data sources was added to the appendix (cf. Fig A8) and a short discussion of these findings has been added. |
| | **Addition to text: "Despite the differences in flow velocities between our maps and the operational products, there is good agreement between the different products on the direction along the flowline (calculated from the X- and Y-Offsets; cf. Fig. A8)."** |

| 1.6 | *"[…] A concluding section on the final error estimate of the ice velocity maps, integrating the outcome of all the independent estimates, is missing."* |
|---|---|
|  | A section including final error estimates has been added to the end of Sect. 4.2. |

| 1.7 | *"Pg. 9 Ln 8: "which corresponds to an area of about 25×25 m." Considering the chosen template size used for the tracking it actually corresponds to a much larger area."* |
|---|---|
|  | After downsampling the offsets to 100x100 m, we calculated the offsets based on the Sentinel-1A scenes on a single pixel basis, not as an average. This sentence has therefore now been deleted. |

| 1.8 | *"Pg. 16 Ln 10: Left uncorrected, these introduce biases in the estimated magnitude of surface velocities (Nagler et al., 2015). ". Did the authors apply any such corrections? If not, what is the estimated bias introduced by this?"* |
|---|---|
|  | As we calculated 2D velocities, we did not apply this type of correction. Given the surface lowering rates of 6 ma$^{-1}$ close to Eqip Sermia's calving front and 2 ma$^{-1}$ at 17 km from the terminus when using TanDEM-X elevation data as a reference (acquisition date 2012/06), the surface slope change amounted to 0.15 degrees in 9 years. This value is almost identical to the one reported for Jakobshavn Glacier by Nagler et al. (2015), reporting a bias of 0.5% in the magnitude of surface velocity caused by surface lowering. These findings have been included in Sect. 4.2. |
|  | **Addition to text: "Based on a comparison between the GIMP DEM and the TanDEM-X 90 m DEM (acquisition date 2012/06;Rizzoli et al., 2017), the surface lowering rates were 6 ma$^{-1}$close to Eqip Sermia's calving front and 2 ma$^{-1}$ at 17 km from the terminus, amounting to a maximum of ~70 m of horizontal location error or less than one pixel in our product. Changes in slope and shape of the glacier need to be accounted for as well when comparing three-dimensional flow magnitudes or flow velocities assuming surface parallel flow, as they can introduce biases in the estimated magnitude of surface velocities (Nagler et al., 2015). Using the calculated lowering rates stated above, this results in a surface slope change of 0.15 degrees in 9 years. This value is almost identical to the one reported for Jakobshavn Glacier by Nagler et al. (2015), reporting a bias of 0.5% in the magnitude of surface velocity caused by surface lowering."** |

| 1.9 | *"Section 4.2: A large part of the discussion concerns an intercomparison of an annually averaged version of the '5m' maps with products from MEaSUREs and CCI. The authors find substantial differences near the calving front and margins but also further upstream. These differences are reported as an underestimation of ice flow in the operational products and according to the authors this implies an underestimation of ice fluxes and an overestimation of Greenland mass loss when used for mass balance assessments. These claims are, however, unsubstantiated by the current work for a number of reasons, aside from the issue regarding resolution described before.:* |
|---|---|
|  | *- Firstly, the reported differences might as well imply an overestimation of the annual product. How is this distinguished? Although the authors mention their claim is supported by the intercomparisons (GPS, UAV, TRI), these only involve short term intercomparisons during a number of episodes in summer when ice flow is usually faster than the annual mean (Pg. 16 ln 17). Also, looking in closer detail at the GPS intercomparison (section 3.2), there appears to be a systematic underestimation for most of the data points (Figure 5). In particular with Sentinel-* |

*1 there are large differences, that seem much higher than the reported 8.7% and are up to nearly 40% (Table 3). In contrast, NASA MEaSUREs reports much better agreement with in-situ GPS (Joughin et al., 2017, 2018).*

- *Secondly, even if for this medium-sized outlet glacier ice flow is underestimated in standardized products, this cannot be generalized into a systematic and substantial underestimation of Greenland ice fluxes, as the authors assert (Pg. 17 Ln 21-30). The nuance seems to be missing here and it appears the paper is overreaching while downplaying uncertainties.*
- *Thirdly an underestimation of flow velocities on an outlet glacier would indeed lead to an underestimation of ice fluxes, but this would also lead to an underestimation of mass loss in the mass budget calculation as less ice is exported. It is unclear why or how this would lead to an overestimation of mass loss as stated by the authors.*

*In order to clarify the discrepancies addressed above, it is necessary to better explain the methods and assumptions used, and to check and revise the error estimates. It could be that perhaps different components are compared (e.g. surface parallel velocity vs horizontal velocity). Also, perhaps the authors are not aware that 250m CCI products are available (Nagler et al., 2015)."*

We refer in the main text to the annual products, as only these allow for an intercomparison of all three glacier-wide products with the same temporal baseline. We agree that the TRI and UAV based flow velocities only represent a snapshot during the summer months, and therefore we added an additional 'short-term' comparison between the MEaSUREs product from August 2016 (monthly velocities) with our SAR based product (see Fig. A3, A5 and A7). The results show a similar characteristic as in the annual comparison, where the MEaSUREs August flow velocities within the last 2.5 km to the glacier front are systematically lower than our estimates, from both Sentinel-1A data and on high resolution and shorter period UAV mosaics (cf. Fig. A3). Given the good agreement between the flow velocities based on UAV data and the interferometrically derived flow velocities from the TRI, we therefore have high confidence in our claim regarding the underestimation of frontal velocities by the operational products.

Given the statement in the abstract regarding the implications due to the underestimation of glacier flow in operational products, we were perhaps overreaching and therefore differentiate this point a bit in the revised text (abstract and discussion). Nevertheless, given the parameters used to produce the operational products (i.e. template window size, sampling step size, ground sampling distance), we think similar differences may occur in terminus flow velocities for outlet glaciers of similar size. We therefore added and specified in the abstract and discussion that this underestimation likely also occurs for other similar sized glaciers.

The passage regarding the underestimation (or as written by us, the overestimation) was incorrect from our side. Thank you for pointing it out, we corrected it accordingly. Furthermore, 250m Greenland Glacier CCI products are available, albeit only for selected glaciers. Greenland-wide products at 250 m are available from ENVEO's Cryoportal, showing significant differences (>1 m) in flow velocities to the 500 m products retrieved from Greenland CCI based on horizontal flow magnitude between 2015/10/01 and 2016/10/31. The differences are apparent along all outlet glaciers and seem to have higher values in the CCI product on the northern glacier margin and higher values in the product from ENVEO's Cryoportal on the glacier's southern margin (see difference map in supplements). Meanwhile, the differences between

| | the 500 m Greenland CCI product and the one retrieved from MEaSUREs with a ground sampling distance of 200 m does not seem to suffer from these effects. It is for this reason that we did not use this 250 m dataset for our analysis. |
|---|---|
| | **Addition to text:**
- **In discussion: "These findings are also valid when comparing the monthly MEaSUREs product from August 2016 with time-averages based on our Sentinel-1A image pairs (cf. Fig A2-A7)."**
- **In discussion: "Given the parameter settings used to produce the operational products (i.e. template window size, sampling step size, ground sampling distance), this underestimation in ice flow near the terminus, may likely apply also to other similar medium-sized outlet glaciers and hence have an impact on mass loss estimates of the whole Greenland Ice Sheet."**
**Changes to text:**
- **In abstract: "[…] which has substantial implications on ice fluxes and on mass budget estimates of similar sized outlet glaciers."**
- **In discussion: "[...] and an underestimation in flow then systematically underestimates mass loss."** |

| 1.10 | *"In several places throughout the manuscript the authors claim their velocity maps as 'improved' over existing products. I have no doubt that these products can be improved in several meaningful ways, including by increasing the spatial/temporal resolution or for example by correction of ionospheric streaks. But, aside from the 'increased' spatial resolution there does not seem to be any further methodological improvement to warrant this claim. Concerning temporal resolution, the CCI project has also provided time series at high temporal resolution, with temporal sampling up to every 6 days, albeit only for selected glaciers (see: http://esa-icesheets-greenland-cci.org). The presented study provides only 12-day maps."* |
|---|---|
| | We adjusted the text accordingly, stressing the importance of small template window sizes in order to be able to resolve the velocity gradients present close to the glacier's terminus (cf. Sect. 4.1 and response to comment 1.1). It is in this regard that our results can be looked at as "improved" over the ice sheet wide, operational products such as MEaSUREs or Greenland Ice Sheet CCI. As pointed out correctly, there are products available at a higher temporal resolution as well, which, unfortunately, is not the case for Eqip Sermia, forcing us to use monthly and yearly data sets for the comparison. Other high-resolution datasets (e.g. TerraSAR-X) are also only available for certain locations and at limited times.
Note that Sentinel-1B completed its commissioning phase only in September 2016, after the field campaign, so the minimal temporal baseline during the campaign was 12 days. |

Figures

| 1.11 | *"Figs 4, 6 & 8: Use same colour scale. The high density of the flow vectors in fig 4 & 6 obscures the velocity map."* |
|---|---|
| | The arrow density and color scale has been adjusted to allow easier reading of colors and to improve comparability of the figures (cf. Fig. 4 & 6) |

| 1.12 | *"Fig 9: Most of the data points seem to lie outside of the glacier (on bedrock?), is this data included in the intercomparison?"* |
|------|------|
|      | No, only the data points inside the glacier's margin (black line in Fig. 9) were included in the intercomparison. The caption has been adjusted to make this clear. |
|      | **Addition to caption: "[…] Only values inside the glacier's marginal boundary (black line) were included in the histogram. […]"** |

| 1.13 | *"Figs 10-15: Difficult to distinguish between the red and orange lines, I would suggest using green instead."* |
|------|------|
|      | Thank you for the suggestion. All orange lines have been changed to green for improved distinguishability. |

| 1.14 | *"Fig 14 & 15: The glacier margins where velocity goes down to zero are missing in this plot. These shear margins are good areas to show the improvement of the increased resolution"* |
|------|------|
|      | The recalculated and newly generated figures for the cross-profiles now show the glacier margins and solid ground, where flow velocities are expected to be low/zero. |

| 1.15 | *"Figs 14 & 15: The x-axis label mentions 'Distance from start of flowline', I assume 'start of profile' is mentioned as this concerns a cross profile."* |
|------|------|
|      | Thank you for pointing this out, has now been corrected in all of the relevant figures. |

**REFEREE COMMENTS #2**

General Comments

| 2.1 | *"[…] The processing chain used to derive ice velocity from the satellite images […] lacks detailed description, and then affects the majority of the results in this paper."* |
|---|---|
| | Additional explanations with further details on the methods have been added (see response 1.4) and the issue about resolution and the downsampling to 100 m added (see reply to comment 1.1) and added further references (see reply to comment 2.4 below) |

Specific Comments

| 2.2 | *"Pg. 1 Ln 21-23: Any reference?"* |
|---|---|
| | The passage is based on information from the paper by Straneo et al. (2013), which was referenced one sentence later. To better clarify the origin of the information, the citation has now been shifted accordingly was moved. |
| | **Changes to text: "As a result of the general warming trend in Greenland and the migration of subtropical water currents toward Greenland's coast, ice loss by submarine melt and iceberg calving – a process neither well understood nor well represented in the current generation of ice-sheet models – is increasing (Straneo et al., 2013). The related dynamic mass loss is expected to further intensify in the future, thereby strongly contributing to global sea level rise (IPCC, 2013; Nick et al., 2013)."** |

| 2.3 | *"Pg. 2 Ln 7-8: A lot of efforts have been made to derive velocity with optical instruments."* |
|---|---|
| | We fully acknowledge the presence of velocity products derived from optical sensors, nevertheless due to the limitations stated on page 2, lines 7-9, it can not be used as a reliable source for operational, year-round observations of flow dynamics at high temporal resolution. We clarified the sentence in this regard. |
| | **Changes to text: "In addition, the operational use of optical remote sensing to measure flow dynamics is limited by the availability of sunlight during the long polar winter as well as cloud cover."** |

| 2.4 | *"Pg. 2 Ln 13-15: Add some more details about the Sentinel-1 mission, or at least some more references."* |
|---|---|
| | We added literature containing further details about the mission and instrument to Sect. 2.1. |
| | **Reference added: Torres, R., Snoeij, P., Geudtner, D., Bibby, D., Davidson, M., Attema, E., Potin, P., Rommen, B., Floury, N., Brown, M., Traver, I. Navas, Deghaye, P., Duesmann, B., Rosich, B., Miranda, N., Bruno, C., L'Abbate, M., Croci, R., Pietropaolo, A., Huchler, M. and Rostan, F.: GMES Sentinel-1 mission, Remote Sensing of Environment, 120, 9–24, doi: 10.1016/j.rse.2011.05.028, 2012.** |

| 2.5 | *"Pg. 3 Ln 7: "high-resolution": Spatial? Temporal? Both?"* |
|---|---|
| | In this passage, we were referring to the spatial dimension, but were not clear enough. We have now adjusted the passage to make this clearer. |

| | **Changes to text: "we demonstrate that flow velocity estimates at a relatively small ground sampling distance are more accurate close to the glacier's terminus compared to operational, ice-sheet wide ice velocity products […]".** |
|---|---|

| 2.6 | *"Pg. 3 Ln 14-17: It is hard to follow. Rewrite the sentence please."* |
|---|---|
| | The sentence was rewritten to make it easier to follow. |
| | **Changes to text: "The long-term flow speed at the terminus was stable for almost a century at about 3 md$^{-1}$ (Bauer, 1968), followed by an acceleration towards the end of the 20$^{th}$ century. Between 2000 and 2005, Eqip Sermia accelerated by 30% as well, doubling the discharge (Rignot and Kanagaratnam, 2006; Kadded and Moreau, 2013; Lüthi et al., 2016)."** |

| 2.7 | *"Pg. 3 Ln 29: "RS-2" was not defined before."* |
|---|---|
| | Added definition to first occurrence. |
| | **Changes to text: "[…] a total of 20 RADARSAT-2 (RS-2) acquisitions were made available […]" "[…] two SLC scenes were acquired using RS-2's Ultra-Fine wide mode […]"** |

| 2.8 | *"Pg. 4 Ln 5: Which version of GIMP do you use? 30m, 90m? How does the DEM oversampling affect the uncertainty in the velocity products?"* |
|---|---|
| | The original resolution of the DEM used was and is 30 m. This information was missing in the manuscript and has been added to section 2.1 (cf. reply to comment 1.4). Since we changed the ground sampling distance of our product to 100x100 m, the DEM oversampling does not affect the velocity product's uncertainty. |
| | **Changes to text: "[…] based on the "GIMP" Digital Elevation Model (DEM). The Greenland Ice Mapping Project (GIMP) DEM has a grid spacing of 30×30 m (Version 2.1; Howat et al., 2014), which was oversampled to 2.5×2.5 m."** |

| 2.9 | *"Pg. 5 Ln 19-21: Reword this sentence. Looks like meteorological conditions does not affect the velocity estimation, which is not true."* |
|---|---|
| | The ambiguous sentence was rewritten, emphasizing that changes in phase information do not have an influence on the amplitude based offset tracking. |
| | **Changes to text: "As this approach does not rely on phase information, using instead the detected SAR image, phase decorrelation caused by meteorological conditions or incoherent and/or rapid flow does not influence the velocity estimation and therefore allows retrievals at higher ice speeds and longer orbit repeat intervals (Gray et al., 2001; Strozzi et al., 2002)."** |

| 2.10 | *"Pg. 5 Ln 6-7: As pointed by Reviewer 1, how does your velocity results have resolution of 5x5m if S1 spatial resolution is 5x20m?"* |
|---|---|
| | (See also reply to comment 1.1) The S1 product was resampled in azimuth to a sample interval of 2.5 m in order to be able to work with "square" pixels while keeping the high resolution in range. This is explained in the section on methods. The velocity products were initially calculated at a 2-pixel sampling step in both directions, resulting in a sampling interval of 5x5 m as originally described in Pg. 5, Ln. 6 f. To account for the correlated data points, we later downsample all offset data sets to a ground sampling distance of 100x100 m. All figures and statistics were recalculated |

| | and the corresponding values in the text were adjusted accordingly (cf. reply to comment 1.1). |
|---|---|

| 2.11 | *"Pg. 5 Ln 9-12: From where, and how long is this "long-term flow velocity average product" ?"* |
|---|---|
| | The long-term flow velocity average product was calculated based on 256 image pairs available between 2014/10/11 and 2018/03/18 using the same offset tracking methodology presented in Sect. 2.2. To calculate the offsets, a template patch size of 101x101 pixels (252.5x252.5 m) was used in combination with search windows of 141x141 pixels, 181x181 pixels, or 261x261 pixels, depending on the temporal baseline of subsequent acquisitions (6, 12, or 24 days). In case of longer temporal baselines (in rare cases of missing acquisitions), no offsets were calculated and thus discarded for the long-term flow velocity average product.
The original manuscript was missing a differentiation of the search window sizes for temporal baselines of 6 and 24 days. This has now been added to Table 2 and the text has been changed to specify this. |
| | **Changes to text: "For this $V_{med}$, offsets in X- and Y-direction were calculated using a 101×101 pixel (252.5×252.5 m) template patch size and search window sizes based on the temporal baseline of each of the 256 image pairs available between 2014/10/11 and 2018/03/18 (cf. Table 2)."** |

| 2.12 | *"Pg. 7 Ln 10-11: Please give a brief description of the three-step approach."* |
|---|---|
| | We added a short description of these steps to Sect. 2.3. |
| | **Additions to text: "[…] using a three-step approach that begins with the culling of outliers, followed by temporally averaging both the latitudinal and longitudinal positions as well as the resulting velocities, in the manner described by Ahlstrøm et al. (2013)."** |

| 2.13 | *"Pg. 9 Figure 4: It's hard to see the colour range on the map behind the arrows. The arrows are great, however there is no scale and they are too small. Please add a scale bar, just to make it easier to the reader."* |
|---|---|
| | Since the color already carries information about the velocity, we decided not to include a reference scale for the arrow length, but to simply include the arrows as a directional reference. In order to improve the readability of the map (especially regarding the velocity information), the arrow density has been decreased. |

| 2.14 | *"Pg. 10 Ln 8: The date 2016/08/13. Typo?"* |
|---|---|
| | Thank you for pointing out this mistake, the correct date should have been 2016/08/21. The typo has been corrected. |
| | **Changes to text: "The UAV-derived velocity field (2016/08/21 and 2016/08/25, Table 3) […]"** |

| 2.15 | *"Pg. 11 Figure 6: Same as Figure 4. It's hard to see the colour range on the map behind the arrows. The arrows are great, however there is no scale and they are too small. Please add a scale bar, just to make it easier to the reader."* |
|---|---|
| | This is accounted for in the response to comment 2.13. |

| 2.16 | *"Pg. 12 Figure 7a: I'd include the Sentinel 2 image in the background as a reference for the masked areas."* |
|---|---|
| | In response, we tested inclusion of the Sentinel 2 image in the background. Unfortunately, the readability of the map became considerably worse due to the relatively small number of overlapping pixels between TRI and the UAV in combination with the bright colors occurring around zero in diverging colormaps. Therefore, we decided to keep the plot as is. |

| 2.17 | *"Pg. 14 Figure 9a: Again, it's very hard to picture the excluded areas."* |
|---|---|
| | See response to 2.16 |

| 2.18 | *"Pg. 16 Ln 9-10: Again, what is the spatial resolution of the DEM?"* |
|---|---|
| | The DEM's original resolution was 30x30 m, oversampled to 2.5x2.5 m (see response to comment 2.8). This information was added to the text. |
| | **Addition to text: "[...] with a nominal date of 2007 and an original resolution of 30×30 m, oversampled to 2.5×2.5 m."** |

| 2.19 | *"Pg. 16 Ln 3-5: "Due to the large spatial coverage […]". I suggest the authors to rewrite this sentence. NASA MEaSURES and ESA CCI are the two main operational centres that deliver velocity products. If the authors look carefully, they do provide high temporal velocity products to specific glaciers."* |
|---|---|
| | Thank you for this suggestion. This was covered in the response to comment 1.9 and 1.10. To point out the existence of products at a high temporal resolution for specific glaciers, we changed the text accordingly. |
| | **Changes to text: "these products are only available for specific glaciers and for specific times at a high temporal resolution and do not cover Eqip Sermia."** |

| 2.20 | *"Pg. 17 Ln 21-24: I suggest the authors to rewrite the sentence. It presents a strong argument, taking in account that this work only provides measurements of a couple of seasons."* |
|---|---|
| | We adjusted the text to emphasize the time period for which the comparison was made (2015/10/01-2016/10/31) in the text. |
| | **Changes to text: "The above differences, calculated for the period between 2015/10/01 and 2016/10/31, imply […]"** |

| 2.21 | *"Pg. 20: "Finally, we were able to demonstrate...": Other studies have already demonstrated the ability of Sentinel-1 to estimate ice velocity near the ice front (e.g. Nagler et al., 2015; Joughin et al, 2018; Lemos et al., 2018). These studies also demonstrated improved results using shorter temporal baseline provided by Sentinel-1 (6 days) since Oct/2016, and the potential of Sentinel-1 to extend existent ice velocity time-series."* |
|---|---|
| | We are aware of these publications that give valuable insights regarding the differences between operational products from different sensors (i.e. TSX, RADARSAT, and PALSAR). However, the focus of our study was on validating the flow speed derivation using S1 amplitude data in close proximity to the calving front using independent measurements from UAV, GPS and TRI. Furthermore, we wanted to raise awareness about the influence of the chosen parameters (e.g. template patch size, sample step |

[revised manuscript text omitted]

**Magnitude Difference**
GL Ice Sheet CCI (500 m) - ENVEO Cryoportal (250 m)
2015/10/01 - 2016/10/31

27 m/d

-8 m/d

0   15   30        60 km

N

---

## Author Response (AR2)

We would like to thank the reviewers for their feedback and their valuable comments on the revised manuscript. In addition to minor orthographic/grammatical corrections as pointed out by reviewer #1 (see below), we respond to the points raised by reviewer #1:

| |
|---|
| ***Comment #1****: "I do not think that the issue regarding the sampling interval is adequately solved by down-sampling the strongly overlapping template patches in a final step. Nor do I believe that it makes much sense to use a 2-pixel sampling step to "improve the representation of velocity gradients" near the calving front as stated now in the newly added paragraph 4.1. For this, the sampling step is not so much of relevance but rather the patch size."* |
| As stated throughout the paper, we are fully aware of the patch size's influence on the algorithms ability to resolve high velocity gradients close to the glacier's calving margin. While a smaller patch size suffers less from a mixture of e.g. sea and ice pixels, the results are considerably noisier when compared with bigger template patches. It is for this reason that an iterative approach with an increasing patch size was chosen, not only helping to reduce the number of void pixels but also to mitigate the drawbacks described above. Furthermore, the influence of different template sizes was discussed in Sect. 4.3.
While the influence of the chosen step size is smaller than the size of the template, it was still present in our results (c.f. Fig. A9) and should therefore be investigated for validation purposes, despite the considerably increased computational cost. Further, given the high resolution of the reference data used for this paper, ignoring this influencing factor would have resulted in additional inconsistencies when directly comparing results from the multiple sensors and would have increased the uncertainties. |
| |
| ***Comment #2:*** *"Maybe for some particular cases it could make sense, […], but certainly not on an operational basis at the ice sheet scale, as the last paragraph in the conclusions and abstract seem to suggest."* |
| As stated in the introduction and the conclusion, the paper's focus is on validating the offset SAR tracking approach with regard to the high velocity gradients occurring close to a glacier's calving margin. Resolving these gradients is of interest, as they strongly influence the estimation of ice discharge with e.g. a flux-gate approach. It was not our intention to suggest using the chosen configuration of step and template patch size to calculate ice velocities on an operational, ice-sheet wide scale. We simply wanted to create awareness of a possible underestimation of ice discharge when using operational, large-scale products. |

**To clarify both considerations, the following paragraph was added to the end of Sect. 4.1. on page 14 f.:**

"Given this paper's focus on *validation* of velocity estimates and spatial patterns using high-resolution reference data, the increased computational cost caused by a 2x2 pixel sample step was acceptable. For processing at e.g. ice-sheet scale, choosing a coarser sample interval (e.g. 20x20 or 40x40 pixels) at the cost of some detail is advisable."

**Minor changes:**

[revised manuscript text omitted]